# Boosting Perturbed Gradient Ascent for Last-Iterate Convergence in Games

**Kenshi Abe**[1,2]    **Mitsuki Sakamoto**[1]    **Kaito Ariu**[1]    **Atsushi Iwasaki**[2]
[1]CyberAgent    [2]The University of Electro-Communications
{abe_kenshi, sakamoto_mitsuki, kaito_ariu}@cyberagent.co.jp
atsushi.iwasaki@uec.ac.jp

## Abstract

This paper presents a payoff perturbation technique, introducing a strong convexity to players' payoff functions in games. This technique is specifically designed for first-order methods to achieve last-iterate convergence in games where the gradient of the payoff functions is monotone in the strategy profile space, potentially containing additive noise. Although perturbation is known to facilitate the convergence of learning algorithms, the magnitude of perturbation requires careful adjustment to ensure last-iterate convergence. Previous studies have proposed a scheme in which the magnitude is determined by the distance from a periodically re-initialized anchoring or reference strategy. Building upon this, we propose Gradient Ascent with Boosting Payoff Perturbation, which incorporates a novel perturbation into the underlying payoff function, maintaining the periodically re-initializing anchoring strategy scheme. This innovation empowers us to provide faster last-iterate convergence rates against the existing payoff perturbed algorithms, even in the presence of additive noise.

## 1 Introduction

This study considers online learning in monotone games, where the gradient of the payoff function is monotone in the strategy profile space. Monotone games encompassed diverse well-studied games as special instances, such as concave-convex games, zero-sum polymatrix games (Cai & Daskalakis, 2011; Cai et al., 2016), $\lambda$-cocoercive games (Lin et al., 2020), and Cournot competition (Monderer & Shapley, 1996). Due to their wide-ranging applications, there has been growing interest in developing learning algorithms to compute Nash equilibria in monotone games.

Typical learning algorithms such as Gradient Ascent (Zinkevich, 2003) and Multiplicative Weights Update (Bailey & Piliouras, 2018) have been extensively studied and shown to converge to equilibria in an average-iterate sense, which is termed *average-iterate convergence*. However, averaging the strategies can be undesirable because it can lead to additional memory or computational costs in the context of training Generative Adversarial Networks (Goodfellow et al., 2014) and preference-based fine-tuning of large language models (Munos et al., 2024; Swamy et al., 2024). In contrast, *last-iterate convergence*, in which the updated strategy profile itself converges to a Nash equilibrium, has emerged as a stronger notion than average-iterate convergence.

Payoff-perturbed algorithms have recently been regaining attention in this context (Sokota et al., 2023; Liu et al., 2023). Payoff perturbation, a classical technique referenced in Facchinei & Pang (2003), introduces a strongly convex penalty to the players' payoff functions to stabilize learning. This leads to convergence toward approximate equilibria, not only in the *full feedback* setting where the perfect gradient vector of the payoff function can be used to update strategies, but also in the *noisy feedback* setting where the gradient vector is contaminated by noise.

However, to ensure convergence toward a Nash equilibrium of the underlying game, the magnitude of perturbation requires careful adjustment. As a remedy, it is adjusted by the distance from an anchoring or reference strategy. Koshal et al. (2010) and Tatarenko & Kamgarpour (2019) simply decay the magnitude in each iteration, and their methods asymptotically converge, since the perturbed function gradually loses strong convexity. In response to this, recent studies (Perolat et al., 2021; Abe et al., 2023; 2024) re-initialize the anchoring strategies periodically, or in a predefined

interval, so that they keep the perturbed function strongly convex and achieve non-asymptotic convergence.

We should also mention the *optimistic* family of learning algorithms, which incorporates recency bias and exhibits last-iterate convergence (Daskalakis et al., 2018; Daskalakis & Panageas, 2019; Mertikopoulos et al., 2019; Wei et al., 2021). Unfortunately, the property has mainly been proven in the full feedback setting. Although it might empirically work with noisy feedback, the convergence is slower, as demonstrated in Section 6. The fast convergence in the noisy feedback setting is another reason why payoff-perturbed algorithms have been gaining renewed interest.

The most recent payoff-perturbed algorithm, *Adaptively Perturbed Mirror Descent* (APMD) (Abe et al., 2024), achieves $\tilde{\mathcal{O}}(1/\sqrt{T})$[1] and $\tilde{\mathcal{O}}(1/T^{\frac{1}{10}})$ last-iterate convergence rates in the full/noisy feedback setting, respectively. The motivation of this study lies in improving these convergence rates. We propose an elegant one-line modification of APMD, which effectively accelerates convergence. In fact, we just add the difference between the current anchoring strategy and the initial anchoring strategy to the payoff perturbation function in APMD.

Our contributions are manifold. Firstly, we propose a novel payoff-perturbed learning algorithm named *Gradient Ascent with Boosting Payoff Perturbation*[2] (GABP). This method incorporates a unique perturbation payoff function, enabling it to achieve fast convergence. Subsequently, we prove that GABP exhibits accelerated $\tilde{\mathcal{O}}(1/T)$ and $\tilde{\mathcal{O}}(1/T^{\frac{1}{7}})$ last-iterate convergence rates to a Nash equilibrium with full and noisy feedback, respectively[3]. To derive these rates, we utilize the concept of the potential function used in Cai & Zheng (2023). Specifically, the potential function we employ is customized for handling noisy feedback. We further show that each player's individual regret is at most $\mathcal{O}\left((\ln T)^2\right)$ in the full feedback setting, provided all players play according to GABP. Finally, through our experiments, we demonstrate the competitive or superior performance of GABP over the Optimistic Gradient algorithm (Daskalakis et al., 2018; Wei et al., 2021), the Accelerated Optimistic Gradient algorithm (Cai & Zheng, 2023), and APMD in concave-convex games, irrespective of the presence of noise.

## 2 PRELIMINARIES

**Monotone games.** In this study, we focus on a continuous multi-player game, which is denoted as $\left([N], (\mathcal{X}_i)_{i\in[N]}, (v_i)_{i\in[N]}\right)$. $[N] = \{1, 2, \cdots, N\}$ denotes the set of $N$ players. Each player $i \in [N]$ chooses a *strategy* $\pi_i$ from a $d_i$-dimensional compact convex strategy space $\mathcal{X}_i$, and we write $\mathcal{X} = \prod_{i\in[N]} \mathcal{X}_i$. Each player $i$ aims to maximize her payoff function $v_i : \mathcal{X} \to \mathbb{R}$, which is differentiable on $\mathcal{X}$. We denote $\pi_{-i} \in \prod_{j\neq i} \mathcal{X}_j$ as the strategies of all players except player $i$, and $\pi = (\pi_i)_{i\in[N]} \in \mathcal{X}$ as the *strategy profile*. This paper particularly studies learning in *smooth monotone games*, where the gradient operator $V(\cdot) = (\nabla_{\pi_i} v_i(\cdot))_{i\in[N]}$ of the payoff functions is monotone: $\forall \pi, \pi' \in \mathcal{X}$,

$$\langle V(\pi) - V(\pi'), \pi - \pi' \rangle \leq 0, \tag{1}$$

and $L$-Lipschitz for $L > 0$,

$$\|V(\pi) - V(\pi')\| \leq L \|\pi - \pi'\|, \tag{2}$$

where $\|\cdot\|$ denotes the $\ell_2$-norm.

Many common and well-studied games, such as concave-convex games, zero-sum polymatrix games (Cai et al., 2016), $\lambda$-cocoercive games (Lin et al., 2020), and Cournot competition (Monderer & Shapley, 1996), are included in the class of monotone games.

**Example 2.1** (Concave-convex games)**.** Consider a game defined by $(\{1, 2\}, (\mathcal{X}_1, \mathcal{X}_2), (v, -v))$, where $v : \mathcal{X}_1 \times \mathcal{X}_2 \to \mathbb{R}$ is the payoff function. In this game, player 1 wishes to maximize $v$, while player 2 aims to minimize $v$. If $v$ is concave in $\pi_1 \in \mathcal{X}_1$ and convex in $\pi_2 \in \mathcal{X}_2$, the game is called a concave-convex game or minimax optimization problem, and it is not hard to see that this game is a special case of monotone games.

---

[1] We use $\tilde{\mathcal{O}}$ to denote a Landau notation that disregards a polylogarithmic factor.

[2] An implementation of our method is available at https://github.com/CyberAgentAILab/boosting-perturbed-ga

[3] For a more detailed comparison of our rates with other works, please refer to Table 3 in Appendix F.2.

**Example 2.2** (Cournot Competition). Consider a Cournot competition model with a linear price function. There are $N$ firms in competition, and each independently and simultaneously chooses a quantity $\pi_i \in \mathcal{X}_i := [0, C_i]$ to produce certain goods, where $C_i$ is a constant greater than 0. The price of the goods is determined by a linear function $P(\pi) = a - b \sum_{i \in [N]} \pi_i$, where $a$ and $b$ are constants greater than 0. The payoff for each firm $i$ is calculated as the total revenue from producing $\pi_i$ units of the goods, minus the associated production cost, i.e., $v_i(\pi) = \pi_i P(\pi) - c_i \pi_i$. This game has been shown to satisfy the property of monotone games as defined in Eq. (1) (Bravo et al., 2018).

**Nash equilibrium and gap function.** A *Nash equilibrium* (Nash, 1951) is a widely used solution concept for a game, which is a strategy profile where no player can gain by changing her own strategy. Formally, a strategy profile $\pi^* \in \mathcal{X}$ is called a Nash equilibrium, if and only if $\pi^*$ satisfies the following condition:

$$\forall i \in [N], \forall \pi_i \in \mathcal{X}_i, \ v_i(\pi_i^*, \pi_{-i}^*) \geq v_i(\pi_i, \pi_{-i}^*).$$

We define the set of all Nash equilibria to be $\Pi^*$. It has been shown that there exists at least one Nash equilibrium (Debreu, 1952) for any smooth monotone games.

To quantify the proximity to Nash equilibrium for a given strategy profile $\pi \in \mathcal{X}$, we use the *gap function*, which is defined as:

$$\mathrm{GAP}(\pi) := \max_{\tilde{\pi} \in \mathcal{X}} \langle V(\pi), \tilde{\pi} - \pi \rangle.$$

Additionally, we use another measure of proximity to Nash equilibrium, referred to as the *tangent residual*. This measure is defined as:

$$r^{\mathrm{tan}}(\pi) := \min_{a \in N_{\mathcal{X}}(\pi)} \|-V(\pi) + a\|,$$

where $N_{\mathcal{X}}(\pi) = \{(a_i)_{i \in [N]} \in \prod_{i=1}^{N} \mathbb{R}^{d_i} \mid \sum_{i=1}^{N} \langle a_i, \pi_i' - \pi_i \rangle \leq 0, \ \forall \pi' \in \mathcal{X}\}$ is the normal cone at $\pi \in \mathcal{X}$. It is easy to see that $\mathrm{GAP}(\pi) \geq 0$ (resp. $r^{\mathrm{tan}}(\pi) \geq 0$) for any $\pi \in \mathcal{X}$, and the equality holds if and only if $\pi$ is a Nash equilibrium. Defining $D := \sup_{\pi, \pi' \in \mathcal{X}} \|\pi - \pi'\|$ as the diameter of $\mathcal{X}$, the gap function for any given strategy profile $\pi \in \mathcal{X}$ is upper bounded by its tangent residual:

**Lemma 2.3** (Lemma 2 of Cai et al. (2022a)). *For any $\pi \in \mathcal{X}$, we have:*

$$\mathrm{GAP}(\pi) \leq D \cdot r^{\mathrm{tan}}(\pi).$$

The gap function and the tangent residual are standard measures of proximity to Nash equilibrium; e.g., it has been used in Cai & Zheng (2023); Abe et al. (2024).

**Problem setting.** This study focuses on the online learning setting in which the following process repeats from iterations $t = 1$ to $T$: (i) Each player $i \in [N]$ chooses her strategy $\pi_i^t \in \mathcal{X}_i$, based on previously observed feedback; (ii) Each player $i$ receives the (noisy) gradient vector $\widehat{\nabla}_{\pi_i} v_i(\pi^t)$ as feedback. This study examines two feedback models: *full feedback* and *noisy feedback*. In the full feedback setting, each player observes the perfect gradient vector $\widehat{\nabla}_{\pi_i} v_i(\pi^t) = \nabla_{\pi_i} v_i(\pi^t)$. In the noisy feedback setting, each player's gradient feedback $\nabla_{\pi_i} v_i(\pi^t)$ is contaminated by an additive noise vector $\xi_i^t$, i.e., $\widehat{\nabla}_{\pi_i} v_i(\pi^t) = \nabla_{\pi_i} v_i(\pi^t) + \xi_i^t$, where $\xi_i^t \in \mathbb{R}^{d_i}$. Throughout the paper, we assume that $\xi_i^t$ is the zero-mean and bounded-variance noise vector at each iteration $t$.

**Payoff-perturbed learning algorithms.** To facilitate the convergence in the online learning setting, recent studies have utilized a *payoff perturbation* technique, where payoff functions are perturbed by strongly convex functions (Sokota et al., 2023; Liu et al., 2023; Abe et al., 2022). However, while the addition of these strongly convex functions leads learning algorithms to converge to a stationary point, this stationary point may be significantly distant from a Nash equilibrium. Therefore, the magnitude of perturbation requires careful adjustment. Perolat et al. (2021); Abe et al. (2023; 2024) have introduced a scheme in which the magnitude is determined by the distance (or divergence function) from an anchoring strategy $\sigma_i$, which is periodically re-initialized. Specifically, Adaptively Perturbed Mirror Descent (APMD) (Abe et al., 2024) perturbs each player's payoff function by a strongly convex divergence function $G(\pi_i, \sigma_i) : \mathcal{X}_i \times \mathcal{X}_i \to [0, \infty)$, where the anchoring strategy $\sigma_i$ is periodically replaced by the current strategy $\pi_i^t$ every predefined iterations $T_\sigma$.

---

**Algorithm 1** GABP for player $i$.

---

**Require:** Learning rates $\{\eta_t\}_{t \geq 0}$, perturbation strength $\mu$, update interval $T_\sigma$, initial strategy $\pi_i^1$

1: $k \leftarrow 1$, $\tau \leftarrow 0$
2: $\sigma_i^1 \leftarrow \pi_i^1$
3: **for** $t = 1, 2, \cdots, T$ **do**
4:      Receive the gradient feedback $\widehat{\nabla}_{\pi_i} v_i(\pi^t)$
5:      Update the strategy by

$$\pi_i^{t+1} = \arg \max_{x \in \mathcal{X}_i} \left\{ \eta_t \left\langle \widehat{\nabla}_{\pi_i} v_i(\pi^t) - \mu \frac{\sigma_i^k - \sigma_i^1}{k+1} - \mu \left( \pi_i^t - \sigma_i^k \right), x \right\rangle - \frac{1}{2} \left\| x - \pi_i^t \right\|^2 \right\}$$

6:      $\tau \leftarrow \tau + 1$
7:      **if** $\tau = T_\sigma$ **then**
8:          $k \leftarrow k + 1$, $\tau \leftarrow 0$
9:          $\sigma_i^k \leftarrow \pi_i^{t+1}$
10:     **end if**
11: **end for**

---

Let us denote the number of updates of $\sigma_i$ up to iteration $t$ as $k(t)$, and the anchoring strategy after $k(t)$ updates as $\sigma_i^{k(t)}$. Since $\sigma_i$ is overwritten every $T_\sigma$ iterations, we can write $k(t) = \lfloor (t-1)/T_\sigma \rfloor + 1$ and $\sigma_i^{k(t)} = \pi_i^{T_\sigma(k(t)-1)+1}$. Except for the payoff perturbation and the update of the anchoring strategy, APMD updates each player $i$'s strategy in the same way as standard Mirror Descent algorithms:

$$\pi_i^{t+1} = \arg \max_{x \in \mathcal{X}_i} \left\{ \eta_t \left\langle \widehat{\nabla}_{\pi_i} v_i(\pi^t) - \mu \nabla_{\pi_i} G(\pi_i^t, \sigma_i^{k(t)}), x \right\rangle - D_\psi(x, \pi_i^t) \right\},$$

where $\eta_t$ is the learning rate at iteration $t$, $\mu \in (0, \infty)$ is the *perturbation strength*, and $D_\psi(\pi_i, \pi_i') = \psi(\pi_i) - \psi(\pi_i') - \langle \nabla \psi(\pi_i'), \pi_i - \pi_i' \rangle$ is the Bregman divergence associated with a strictly convex function $\psi : \mathcal{X}_i \to \mathbb{R}$. When both $G$ and $D_\psi$ is set to the squared $\ell^2$-distance, this algorithm can be equivalently written as:

$$\pi_i^{t+1} = \arg \max_{x \in \mathcal{X}_i} \left\{ \eta_t \left\langle \widehat{\nabla}_{\pi_i} v_i(\pi^t) - \mu \left( \pi_i^t - \sigma_i^{k(t)} \right), x \right\rangle - \frac{1}{2} \left\| x - \pi_i^t \right\|^2 \right\}.$$

We refer to this version of APMD as Adaptively Perturbed Gradient Ascent (APGA). Abe et al. (2024) have shown that APGA exhibits the convergence rates of $\tilde{\mathcal{O}}(1/\sqrt{T})$ and $\tilde{\mathcal{O}}(1/T^{\frac{1}{10}})$ with full and noisy feedback, respectively.

## 3 GRADIENT ASCENT WITH BOOSTING PAYOFF PERTURBATION

This section proposes a novel payoff-perturbed learning algorithm, Gradient Ascent with Boosting Payoff Perturbation (GABP). The pseudo-code of GABP is outlined in Algorithm 1. At each iteration $t \in [T]$, GABP receives the gradient feedback $\widehat{\nabla}_{\pi_i} v_i(\pi^t)$, and updates each player's strategy by the following update rule:

$$\pi_i^{t+1} = \arg \max_{x \in \mathcal{X}_i} \left\{ \eta_t \left\langle \widehat{\nabla}_{\pi_i} v_i(\pi^t) - \underbrace{\mu \frac{\sigma_i^{k(t)} - \sigma_i^1}{k(t)+1}}_{(*)} - \mu \left( \pi_i^t - \sigma_i^{k(t)} \right), x \right\rangle - \frac{1}{2} \left\| x - \pi_i^t \right\|^2 \right\}. \quad (3)$$

$\sigma_i$ is overwritten every $T_\sigma$ iterations, and thus $\sigma_i^{k(t)}$ is define as $\sigma_i^{k(t)} = \pi_i^{T_\sigma(k(t)-1)+1}$. The term $(*)$ in Eq. (3) is our proposed additional perturbation term. It shrinks as $k(t)$, the number of updates of $\sigma_i^{k(t)}$, increases.

For a more intuitive explanation of the proposed perturbation term, we present the following update rule, which is equivalent to Eq. (3):

$$\pi_i^{t+1} = \arg\max_{x \in \mathcal{X}_i} \left\{ \eta_t \left\langle \widehat{\nabla}_\pi v_i(\pi^t) - \mu \left( \pi_i^t - \frac{k(t)\sigma_i^{k(t)} + \sigma_i^1}{k(t)+1} \right), x \right\rangle - \frac{1}{2} \left\| x - \pi_i^t \right\|^2 \right\}.$$

According this formula, it is evident that GABP perturbs the gradient vector $\widehat{\nabla}_\pi v_i(\pi^t)$ so that $\pi_i^t$ approaches $\frac{k(t)\sigma_i^{k(t)} + \sigma_i^1}{k(t)+1}$, instead of $\sigma_i^{k(t)}$. This gradual evolution of the anchoring strategy in GABP, compared to $\sigma_i^{k(t)}$ itself, is anticipated to contribute to the stabilization of the learning dynamics. There is a tradeoff between the shrinking speed of the term (∗) and the stabilizing impact on the last-iterate convergence rate of GABP. The shrinking speed of $1/(k(t)+1)$ achieves a faster convergence rate, and we believe that this represents the optimal balance for this trade-off. We remark that the term (∗) bears a resemblance to the update rule of the Accelerated Optimistic Gradient (AOG) algorithm (Cai & Zheng, 2023). However, AOG differs in the sense that it actually modifies the proximal point in gradient ascent, instead of perturbing the gradient vector. A detailed comparison is discussed in Appendix F.1.

## 4    LAST-ITERATE CONVERGENCE RATES

This section provides the last-iterate convergence rates of GABP. Specifically, we derive $\tilde{\mathcal{O}}(1/T)$ and $\tilde{\mathcal{O}}(1/T^{\frac{1}{7}})$ rates for the full/noisy feedback setting, respectively.

### 4.1    FULL FEEDBACK SETTING

First, we demonstrate the last-iterate convergence rate of GABP with *full feedback* where each player receives the perfect gradient vector as feedback at each iteration $t$, i.e., $\widehat{\nabla}_{\pi_i} v_i(\pi^t) = \nabla_{\pi_i} v_i(\pi^t)$. Theorem 4.1 shows that the last-iterate strategy profile $\pi^T$ updated by GABP converges to a Nash equilibrium with an $\tilde{\mathcal{O}}(1/T)$ rate in the full feedback setting.

**Theorem 4.1.** *If we use the constant learning rate $\eta_t = \eta \in (0, \frac{\mu}{(L+\mu)^2})$ and the constant perturbation strength $\mu > 0$, and set $T_\sigma = c \cdot \max(1, \frac{6\ln 3(T+1)}{\ln(1+\eta\mu)})$ for some constant $c \geq 1$, then the strategy $\pi^t$ updated by GABP satisfies for any $t \in [T]$:*

$$\text{GAP}(\pi^{t+1}) \leq D \cdot r^{\tan}(\pi^{t+1}) \leq \frac{17cD^2 \left( \frac{6\ln 3(T+1)}{\ln(1+\eta\mu)} + 1 \right)}{t} \left( \mu + \frac{1+\eta L}{\eta} \right).$$

This rate is competitive compared to the previous state-of-the-art rate of $\mathcal{O}(1/T)$ (Yoon & Ryu, 2021; Cai & Zheng, 2023). Note that the rate in Theorem 4.1 holds for any fixed $\mu > 0$.

### 4.2    NOISY FEEDBACK SETTING

Next, we establish the last-iterate convergence rate in the *noisy feedback* setting, where each player $i$ observes a noisy gradient vector contaminated by an additive noise vector $\xi_i^t \in \mathbb{R}^{d_i}$: $\widehat{\nabla}_{\pi_i} v_i(\pi^t) = \nabla_{\pi_i} v_i(\pi^t) + \xi_i^t$. We assume that the noisy vector $\xi_i^t$ is zero-mean and its variance is bounded. Formally, defining the sigma-algebra generated by the history of the observations as $\mathcal{F}_t := \sigma \left( (\widehat{\nabla}_{\pi_i} v_i(\pi^1))_{i \in [N]}, \ldots, (\widehat{\nabla}_{\pi_i} v_i(\pi^{t-1}))_{i \in [N]} \right), \forall t \geq 1$, the noisy vector $\xi_i^t$ is assumed to satisfy the following conditions:

**Assumption 4.2.** *$\xi_i^t$ satisfies the following properties for all $t \geq 1$ and $i \in [N]$: (a) Zero-mean: $\mathbb{E}[\xi_i^t | \mathcal{F}_t] = (0, \cdots, 0)^\top$; (b) Bounded variance: $\mathbb{E}[\|\xi_i^t\|^2 | \mathcal{F}_t] \leq C^2$ with some constant $C > 0$.*

Assumption 4.2 is standard in online learning in games with noisy feedback (Mertikopoulos & Zhou, 2019; Hsieh et al., 2019; Abe et al., 2024) and stochastic optimization (Nemirovski et al., 2009; Nedić & Lee, 2014). Under Assumption 4.2 and a decreasing learning rate sequence $\eta_t$, we can obtain a faster last convergence rate $\tilde{\mathcal{O}}(1/T^{\frac{1}{7}})$ than $\tilde{\mathcal{O}}(1/T^{\frac{1}{10}})$ of APGA (Abe et al., 2024).

**Theorem 4.3.** *Let $\kappa = \frac{\mu}{2}, \theta = \frac{3\mu^2 + 8L^2}{2\mu}$. Suppose that Assumption 4.2 holds and $V(\pi) \leq \zeta$ for any $\pi \in \mathcal{X}$. We also assume that $T_\sigma$ is set to satisfy $T_\sigma = c \cdot \max(T^{\frac{6}{7}}, 1)$ for some constant $c \geq 1$. If we use the constant perturbation strength $\mu > 0$ and the decreasing learning rate sequence $\eta_t = \frac{1}{\kappa(t - T_\sigma(k(t)-1)) + 2\theta}$, then the strategy $\pi^{T+1}$ satisfies:*

$$\mathbb{E}\left[\mathrm{GAP}(\pi^{T+1})\right] = \mathcal{O}\left(\frac{\ln T}{T^{\frac{1}{7}}}\right).$$

### 4.3 Proof sketch of Theorems 4.1 and 4.3

This section outlines the sketch of the proofs for Theorems 4.1 and 4.3. The complete proofs are placed in Appendix B and C.

We define the stationary point $\pi^{\mu,k(t)}$, which satisfies the following condition: $\forall i \in [N]$,

$$\pi_i^{\mu,k(t)} = \arg\max_{x \in \mathcal{X}_i}\left\{v_i(x, \pi_{-i}^{\mu,k(t)}) - \frac{\mu}{2}\left\|x - \hat{\sigma}_i^{k(t)}\right\|^2\right\},$$

where $\hat{\sigma}_i^{k(t)} = \frac{k(t)\sigma_i^{k(t)} + \sigma_i^1}{k(t)+1}$. The primary technical challenge in deriving the last-iterate convergence rates lies in the construction of the following potential function $P^{k(t)}$, which can be utilized in the proofs for both full and noisy feedback settings:

$$P^{k(t)} := k(t)(k(t)+1)\left(\frac{\left\|\pi^{\mu,k(t)-1} - \hat{\sigma}^{k(t)-1}\right\|^2}{2} + \left\langle \hat{\sigma}^{k(t)} - \pi^{\mu,k(t)-1}, \pi^{\mu,k(t)-1} - \hat{\sigma}^{k(t)-1}\right\rangle\right).$$

Specifically, we demonstrate that this potential function is approximately non-increasing regardless of the presence of noise. Although the potential function $P^{k(t)}$ is inspired by one in Cai & Zheng (2023), their potential function contains the term $\eta^2 \sum_{i\in[N]}\left\|\widehat{\nabla}v_i(\pi^t) - \widehat{\nabla}v_i(\pi^{t-\frac{1}{2}})\right\|^2$, which could have a high value in the noisy feedback setting even if $\pi^t = \pi^{t-\frac{1}{2}}$ holds[4]. This complicates providing a last-iterate convergence result for the noisy feedback setting via their potential function. In contrast, our potential function $P^{k(t)}$ does not include the term dependent on $\widehat{\nabla}v_i(\pi^t)$. As a result, we can provide the last-iterate convergence rates even for the noisy feedback setting.

Furthermore, compared to Abe et al. (2024), our new potential function $P^{k(t)}$ allows us to improve the convergence rate with respect to the distance between the $k(t)$-th anchoring strategy profile $\sigma^{k(t)}$ and the corresponding stationary point $\pi^{\mu,k(t)}$, i.e., $\left\|\pi^{\mu,k(t)} - \sigma^{k(t)}\right\|$. In fact, our analysis enhances this rate from $\mathcal{O}(1/\sqrt{k(t)})$ to $\mathcal{O}(1/k(t))$. This improvement leads to faster last-iterate convergence results, further demonstrating the effectiveness of our approach.

**(1) Potential function for bounding the distance between $\pi^{\mu,k(t)}$ and $\sigma^{k(t)}$.** As mentioned above, our main technical contribution is proving that $P^{k(t)}$ is approximately non-increasing (as shown in Lemma B.3). That is, we have for any $t \geq 1$ such that $k(t) \geq 2$:

$$P^{k(t)+1} \leq P^{k(t)} + (k(t)+1)^2 \cdot \mathcal{O}\left(\left\|\pi^{\mu,k(t)} - \sigma^{k(t)+1}\right\| + \left\|\pi^{\mu,k(t)-1} - \sigma^{k(t)}\right\|\right). \quad (4)$$

By telescoping of Eq. (4) and the first-order optimality condition for $\pi^{\mu,k(t)}$, we can derive the following upper bound on the distance between $\pi^{\mu,k}$ and $\hat{\sigma}^{k(t)}$:

$$\frac{(k(t)+1)(k(t)+2)}{2}\left\|\pi^{\mu,k(t)} - \hat{\sigma}^{k(t)}\right\|^2 \leq \mathcal{O}(1) + (k(t)+1)^2 \sum_{l=1}^{k(t)}\mathcal{O}\left(\left\|\pi^{\mu,l} - \sigma^{l+1}\right\|\right).$$

Applying the definition of $\hat{\sigma}^{k(t)}$ and Cauchy-Schwarz inequality to this inequality, we obtain:

$$\left\|\pi^{\mu,k(t)} - \sigma^{k(t)}\right\|^2 \leq \frac{\mathcal{O}\left(\left\|\pi^{\mu,k(t)} - \sigma^{k(t)}\right\|\right)}{k(t)+1} + \mathcal{O}\left(\frac{1}{(k(t)+1)^2}\right) + \sum_{l=1}^{k(t)}\mathcal{O}\left(\left\|\pi^{\mu,l} - \sigma^{l+1}\right\|\right). \quad (5)$$

---

[4]The comparison of the potential function of Cai & Zheng (2023) with ours can be found in Appendix F.1.

Note that the non-increasing property of our potential function, as described in Eq. (4), holds even in the noisy feedback setting. This implies that a similar proof technique for deriving Eq. (5) can be utilized to provide last-iterate convergence results both in full and noisy feedback settings.

**(2) Convergence rate of $\sigma^{k(t)+1}$ to the stationary point $\pi^{\mu,k(t)}$.** Leveraging the strong convexity of the perturbation payoff function, $\frac{\mu}{2}\|x - \hat{\sigma}_i^{k(t)}\|^2$, we show that $\pi^t$ converges to $\pi^{\mu,k(t)}$ exponentially fast in the full feedback setting (as shown in Lemma B.1). Specifically, we have for any $t \geq 1$:

$$\left\|\pi^{\mu,k(t)} - \pi^{t+1}\right\|^2 \leq \left(\frac{1}{1+\eta\mu}\right)^{t-(k(t)-1)T_\sigma} \left\|\pi^{\mu,k(t)} - \sigma^{k(t)}\right\|^2. \tag{6}$$

By using Eq. (6) and the assumption that $T_\sigma \geq \frac{6\ln 3(T+1)}{\ln(1+\eta\mu)}$ in the full feedback setting, we can easily show that $\left\|\pi^{\mu,l} - \sigma^{l+1}\right\| = \left\|\pi^{\mu,l} - \pi^{T_\sigma l+1}\right\| \leq \mathcal{O}\big(k(t)^{-3}\big)$ for any $l \leq k(t)$. Hence, from Eq. (5), we can derive the following convergence rate of the distance between $\pi^{\mu,k(t)}$ and $\sigma^{k(t)}$ with respect to $k(t)$ (as shown in Lemma B.2):

$$\left\|\pi^{\mu,k(t)} - \sigma^{k(t)}\right\| \leq \mathcal{O}(1/k(t)). \tag{7}$$

**(3) Decomposition of the gap function of the last-iterate strategy profile $\pi^{T+1}$.** Let us define $K := \lfloor T/T_\sigma \rfloor$. From Cauchy-Schwarz inequality and Lemma 2.3, we can decompose the gap function $\mathrm{GAP}(\pi^{T+1})$ as follows:

$$\mathrm{GAP}(\pi^{T+1}) \leq \mathrm{GAP}(\pi^{\mu,K}) + \mathcal{O}\left(\left\|\pi^{\mu,K} - \pi^{T+1}\right\|\right)$$
$$\leq D \cdot \min_{a \in N_\mathcal{X}(\pi^{\mu,K})} \left\|-V(\pi^{\mu,K}) + a\right\| + \mathcal{O}\left(\left\|\pi^{\mu,K} - \pi^{T+1}\right\|\right).$$

From the first-order optimality condition for $\pi^{\mu,K}$, we can see that $V(\pi^{\mu,K}) - \mu(\pi^{\mu,K} - \hat{\sigma}^K) \in N_\mathcal{X}(\pi^{\mu,K})$. Thus, from the triangle inequality and $L$-smoothness of the gradient operator in Eq. (2), the gap function $\mathrm{GAP}(\pi^{T+1})$ can be bounded as:

$$\mathrm{GAP}(\pi^{T+1}) \leq \mathcal{O}(1/K) + \mathcal{O}\left(\left\|\pi^{\mu,K} - \sigma^K\right\|\right) + \mathcal{O}\left(\left\|\pi^{\mu,K} - \pi^{T+1}\right\|\right). \tag{8}$$

**(4) Putting it all together: last-iterate convergence rate of $\pi^{T+1}$.** By combining Eq. (6), Eq. (7), and Eq. (8), it holds that $\mathrm{GAP}(\pi^{T+1}) \leq \mathcal{O}(1/K)$ in the full feedback setting. Hence, given $K = \lfloor T/T_\sigma \rfloor$, we can deduce that $\mathrm{GAP}(\pi^{T+1}) \leq \mathcal{O}(T_\sigma/T)$. Finally, taking $T_\sigma = \Theta(\ln T)$, we obtain the upper bound on the gap function for the full feedback setting: $\mathrm{GAP}(\pi^{T+1}) \leq \mathcal{O}(\ln T/T)$. Note that using a similar proof technique, we can also derive an upper bound on the tangent residual for the full feedback setting.

In the context of the noisy feedback setting, we achieve the following convergence rate to $\pi^{\mu,k(t)}$ instead of Eq. (6) (as shown in Lemma C.1):

$$\mathbb{E}\left[\left\|\pi^{\mu,k(t)} - \pi^{t+1}\right\|^2\right] \leq \mathcal{O}\left(\frac{\ln t}{t - (k(t) - 1)T_\sigma}\right). \tag{9}$$

By using Eq. (9) and the assumption that $T_\sigma = \Theta(T^{\frac{6}{7}})$, we can still derive Eq. (7) and Eq. (8) for the noisy feedback setting. Therefore, we conclude that: $\mathbb{E}\left[\mathrm{GAP}(\pi^{T+1})\right] \leq \mathcal{O}(\ln T/T^{\frac{1}{7}})$.

## 5 INDIVIDUAL REGRET BOUND

In this section, we present an upper bound on an individual regret for each player. Specifically, our study examines two performance measures: the *external regret* and the *dynamic regret* (Zinkevich, 2003). The external regret is a conventional measure in online learning. In online learning in games, the external regret for player $i$ is defined as the gap between the player's realized cumulative payoff and the cumulative payoff of the best fixed strategy in hindsight:

$$\mathrm{Reg}_i(T) := \max_{x \in \mathcal{X}_i} \sum_{t=1}^{T} \left(v_i(x, \pi_{-i}^t) - v_i(\pi^t)\right).$$

The dynamics regret is a much stronger performance metric, which is given by:

$$\text{DynamicReg}_i(T) := \sum_{t=1}^{T} \left( \max_{x \in \mathcal{X}_i} v_i(x, \pi_{-i}^t) - v_i(\pi^t) \right).$$

We show in Theorem 5.1 that the individual regret is at most $\mathcal{O}\left((\ln T)^2\right)$ if each player $i \in [N]$ plays according to GABP in the full feedback setting. The proof is given in Appendix D.

**Theorem 5.1.** *In the same setup of Theorem 4.1, we have for any player $i \in [N]$ and $T \geq 2$:*

$$\text{Reg}_i(T) \leq \text{DynamicReg}_i(T) \leq \mathcal{O}\left((\ln T)^2\right).$$

This regret bound is significantly superior to the $\mathcal{O}(\sqrt{T})$ regret bound of the Optimistic Gradient (OG) algorithm, and it is slightly inferior to the $\mathcal{O}(\ln T)$ regret bound of AOG (Cai & Zheng, 2023).

## 6  EXPERIMENTS

In this section, we present the empirical results of our GABP, comparing its performance with APGA (Abe et al., 2024), OG (Daskalakis et al., 2018; Wei et al., 2021), and AOG (Cai & Zheng, 2023). We conduct experiments on two classes of concave-convex games. One is random payoff games, which are two-player zero-sum normal-form games with payoff matrices of size $d$. Each player's strategy space is represented by the $d$-dimensional probability simplex, i.e., $\mathcal{X}_1 = \mathcal{X}_2 = \Delta^d$. All entries of the payoff matrix are drawn independently from a uniform distribution over the interval $[-1, 1]$. We set $d = 50$ and the initial strategies are set to $\pi_1^1 = \pi_2^1 = \frac{1}{d}\mathbf{1}$. The other is a *hard concave-convex game* (Ouyang & Xu, 2021), formulated as the following max-min optimization problem: $\max_{x \in \mathcal{X}_1} \min_{y \in \mathcal{X}_2} f(x, y)$, where $f(x, y) = -\frac{1}{2}x^\top H x + h^\top x + \langle Ax - b, y \rangle$. Following the setup in Cai & Zheng (2023), we choose $\mathcal{X}_1 = \mathcal{X}_2 = [-200, 200]^d$ with $d = 100$. The precise terms of $H \in \mathbb{R}^{d \times d}, A \in \mathbb{R}^{d \times d}, b \in \mathbb{R}^d$, and $h \in \mathbb{R}^d$ are provided in Appendix E.2. All algorithms are executed with initial strategies $\pi_1^1 = \pi_2^1 = \frac{1}{d}\mathbf{1}$. The detailed hyperparameters of the algorithms, tuned for best performance, are shown in Table 2 in Appendix E.3.

Figure 1 illustrates the logarithmic GAP values per iteration for each of the two games under each type of feedback. For the random payoff games with full or noisy feedback, 50 payoff matrices are generated using different random seeds. Likewise, for the hard concave-convex games, we use 10 different random seeds. We assume that the noise vector $\xi_i^t$ is generated from the multivariate Gaussian distribution $\mathcal{N}(0, \ 0.1^2\mathbf{I})$ in an i.i.d. manner for both games. In the former game with full feedback, GABP performs almost as well as the others. With noisy feedback, GABP outperforms the others, although the margin from APGA is slight. In the latter game, under the full feedback setting, GABP is competitive against AOG, whereas, under the noisy feedback setting, it demonstrates a substantial advantage over the others.

Figure 2 illustrates the dynamic regret in the hard concave-convex game. GABP exhibits lower regret than APGA and OG with both feedback, demonstrating its efficiency and robustness. Note that APGA and OG exhibit almost identical trajectories with full feedback, with their plots overlapping completely. In addition, GABP achieves competitive regret in comparison to AOG.

## 7  RELATED LITERATURE

No-regret learning algorithms have been extensively studied with the intent of achieving key objectives such as average-iterate convergence or last-iterate convergence. Recently, learning algorithms introducing optimism (Rakhlin & Sridharan, 2013a;b), such as optimistic Follow the Regularized Leader (Shalev-Shwartz & Singer, 2006) and optimistic Mirror Descent (Zhou et al., 2017; Hsieh et al., 2021), have been introduced to admit last-iterate convergence in a broad spectrum of game settings. These optimistic algorithms with full feedback have been shown to achieve last-iterate convergence in various classes of games, including bilinear games (Daskalakis et al., 2018; Daskalakis & Panageas, 2019; Liang & Stokes, 2019; de Montbrun & Renault, 2022), cocoercive games (Lin et al., 2020), and saddle point problems (Daskalakis & Panageas, 2018; Mertikopoulos et al., 2019; Golowich et al., 2020b; Wei et al., 2021; Lei et al., 2021; Yoon & Ryu, 2021; Lee & Kim, 2021;

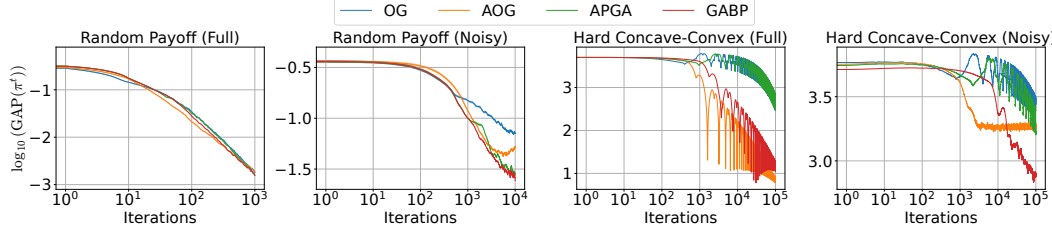

Figure 1: The gap function for $\pi^t$ of GABP, APGA, OG, and AOG with full and noisy feedback. The shaded area represents the standard errors.

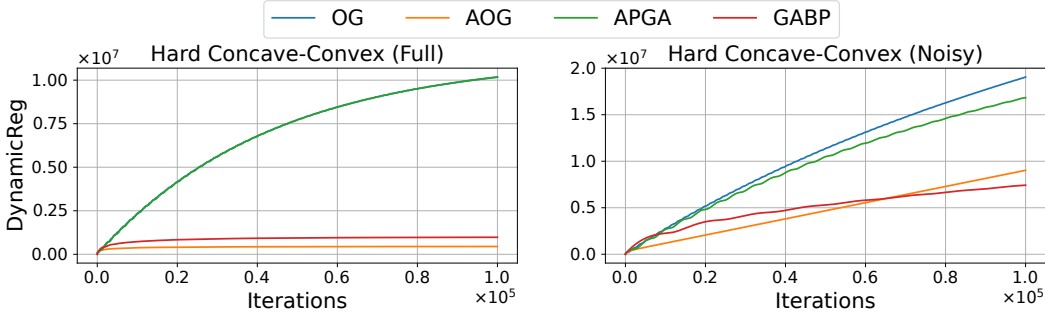

Figure 2: Dynamic regret for GABP, APGA, OG, and AOG with full and noisy feedback.

Cevher et al., 2023). Recent studies have provided finite convergence rates for monotone games (Golowich et al., 2020a; Cai et al., 2022a;b; Gorbunov et al., 2022; Cai & Zheng, 2023).

Compared to the full feedback setting, there are significant challenges in learning with noisy feedback. For example, a learning algorithm must estimate the gradient from feedback that is contaminated by noise. Despite the challenge, a vast literature has successfully achieved last-iterate convergence with noisy feedback in specific classes of games, including potential games (Cohen et al., 2017), strongly monotone games (Giannou et al., 2021b;a), and two-player zero-sum games (Abe et al., 2023). These results have often leveraged unique structures of their payoff functions, such as strict (or strong) monotonicity (Bravo et al., 2018; Kannan & Shanbhag, 2019; Hsieh et al., 2019; Anagnostides & Panageas, 2022) and strict variational stability (Mertikopoulos & Zhou, 2019; Mertikopoulos et al., 2019; 2022; Azizian et al., 2021). Without these restrictions, convergence is mainly demonstrated in an asymptotic manner, with no quantification of the rate (Hsieh et al., 2020; 2022; Abe et al., 2023). Consequently, an exceedingly large number of iterations might be necessary to reach an equilibrium.

There have been several studies focusing on payoff-regularized learning, where each player's payoff or utility function is perturbed or regularized via strongly convex functions (Cen et al., 2021; 2023; Pattathil et al., 2023). Previous studies have successfully achieved convergence to stationary points, which are approximate equilibria. For instance, Sokota et al. (2023) have demonstrated that their perturbed mirror descent algorithm converges to a quantal response equilibrium (McKelvey & Palfrey, 1995; 1998). Similar results have been obtained with the Boltzmann Q-learning dynamics (Tuyls et al., 2006) and penalty-regularized dynamics (Coucheney et al., 2015) in continuous-time settings (Leslie & Collins, 2005; Abe et al., 2022; Hussain et al., 2023). To ensure convergence toward a Nash equilibrium of the underlying game, the magnitude of perturbation requires careful adjustment. Several learning algorithms have been proposed to gradually reduce the perturbation strength $\mu$ in response to this (Bernasconi et al., 2022; Liu et al., 2023; Cai et al., 2023). These include well-studied methods such as iterative Tikhonov regularization (Facchinei & Pang, 2003; Koshal et al., 2010; 2013; Yousefian et al., 2017; Tatarenko & Kamgarpour, 2019). Alternatively, Perolat et al. (2021) and Abe et al. (2023) have employed a payoff perturbation scheme, where the magnitude of perturbation is determined by the distance from an anchoring strategy, which is periodically re-initialized by the current strategy. Recently, Abe et al. (2024) have established $\tilde{\mathcal{O}}(1/\sqrt{T})$

and $\tilde{\mathcal{O}}(1/T^{\frac{1}{10}})$ last-iterate convergence rates for the payoff perturbation scheme in the full/noisy feedback setting, respectively. Our algorithm achieves faster $\tilde{\mathcal{O}}(1/T)$ and $\tilde{\mathcal{O}}(1/T^{\frac{1}{7}})$ last-iterate convergence rates by modifying the periodically re-initializing anchoring strategy scheme so that the anchoring strategy evolves more gradually.

## 8 CONCLUSION

This study proposes a novel payoff-perturbed algorithm, Gradient Ascent with Boosting Payoff Perturbation, which achieves $\tilde{\mathcal{O}}(1/T)$ and $\tilde{\mathcal{O}}(1/T^{\frac{1}{7}})$ last-iterate convergence rates in monotone games with full/noisy feedback, respectively. Extending our results in settings where each player only observes bandit feedback is an intriguing and challenging future direction.

## ACKNOWLEDGMENTS

Kaito Ariu is supported by JSPS KAKENHI Grant Number 23K19986. Atsushi Iwasaki is supported by JSPS KAKENHI Grant Numbers 21H04890 and 23K17547.

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

# A  NOTATIONS

In this section, we summarize the notations we use in Table 1.

Table 1: Notations

| Symbol | Description |
|---|---|
| $N$ | Number of players |
| $\mathcal{X}_i$ | Strategy space for player $i$ |
| $\mathcal{X}$ | Joint strategy space: $\mathcal{X} = \prod_{i=1}^{N} \mathcal{X}_i$ |
| $v_i$ | Payoff function for player $i$ |
| $\pi_i$ | Strategy for player $i$ |
| $\pi$ | Strategy profile: $\pi = (\pi_i)_{i \in [N]}$ |
| $\pi^*$ | Nash equilibrium |
| $\Pi^*$ | Set of Nash equilibria |
| $\mathrm{GAP}(\pi)$ | Gap function of $\pi$: $\mathrm{GAP}(\pi) = \max_{\tilde{\pi} \in \mathcal{X}} \sum_{i=1}^{N} \langle \nabla_{\pi_i} v_i(\pi), \tilde{\pi}_i - \pi_i \rangle$ |
| $r^{\tan}(\pi)$ | Tangent residual of $\pi$: $r^{\tan}(\pi) = \min_{a \in N_{\mathcal{X}}(\pi)} \|-V(\pi) + a\|$ |
| $\nabla_{\pi_i} v_i(\pi)$ | Gradient vector of $v_i$ with respect to $\pi_i$ |
| $\widehat{\nabla}_{\pi_i} v_i(\pi)$ | Noisy gradient vector of $v_i$ with respect to $\pi_i$: $\widehat{\nabla}_{\pi_i} v_i(\pi) = \nabla_{\pi_i}(\pi) + \xi_i^t$ |
| $\xi_i^t$ | Noise vector for player $i$ at iteration $t$ |
| $V(\cdot)$ | Gradient operator of the payoff functions: $V(\cdot) = (\nabla_{\pi_i} v_i(\cdot))_{i \in [N]}$ |
| $T$ | Total number of iterations |
| $\eta_t$ | Learning rate at iteration $t$ |
| $\mu$ | Perturbation strength |
| $T_\sigma$ | Update interval for the anchoring strategy |
| $\pi^t$ | Strategy profile at iteration $t$ |
| $k(t)$ | Number of updates of the anchoring strategy up to iteration $t$ |
| $K$ | Total number of the updates of the anchoring strategy |
| $\sigma^{k(t)}$ | Anchoring strategy profile at iteration $t$ |
| $\hat{\sigma}^{k(t)}$ | convex combination of $\sigma^{k(t)}$ and $\sigma^1$: $\hat{\sigma}^{k(t)} = \frac{k(t)\sigma^{k(t)} + \sigma^1}{k(t)+1}$ |
| $\pi^{\mu,k(t)}$ | Stationary point satisfies: $\forall i \in [N],\ \pi_i^{\mu,k(t)} = \arg\max_{x \in \mathcal{X}_i} \left\{ v_i(x, \pi_{-i}^{\mu,k(t)}) - \frac{\mu}{2} \|x - \hat{\sigma}^{k(t)}\|^2 \right\}$ |
| $L$ | Smoothness parameter of $(v_i)_{i \in [N]}$ |

# B  PROOFS FOR THEOREM 4.1

## B.1  PROOF OF THEOREM 4.1

*Proof of Theorem 4.1.* From the first-order optimality condition for $\pi^t$, we have for any $x \in \mathcal{X}$ and $t \geq 2$:

$$\left\langle V(\pi^{t-1}) - \mu \left( \pi^{t-1} - \frac{k(t-1)\sigma^{k(t-1)} + \sigma^1}{k(t-1)+1} \right) - \frac{1}{\eta} \left( \pi^t - \pi^{t-1} \right), \pi^t - x \right\rangle \geq 0,$$

and then $V(\pi^{t-1}) - \mu \left( \pi^{t-1} - \frac{k(t-1)\sigma^{k(t-1)} + \sigma^1}{k(t-1)+1} \right) - \frac{1}{\eta} \left( \pi^t - \pi^{t-1} \right) \in N_{\mathcal{X}}(\pi^t)$. Thus, the tangent residual for $\pi^t$ can be bounded as:

$$r^{\tan}(\pi^t) = \min_{a \in N_{\mathcal{X}}(\pi^t)} \|-V(\pi^t) + a\|$$

$$\leq \left\| -V(\pi^t) + V(\pi^{t-1}) - \mu \left( \pi^{t-1} - \frac{k(t-1)\sigma^{k(t-1)} + \sigma^1}{k(t-1)+1} \right) - \frac{1}{\eta} \left( \pi^t - \pi^{t-1} \right) \right\|.$$

Letting us define

$$\pi_i^{\mu,k} = \arg\max_{\pi_i \in \mathcal{X}_i} \left\{ v_i(\pi_i, \pi_{-i}^{\mu,k}) - \frac{\mu}{2} \left\| \pi_i - \frac{k\sigma_i^k + \sigma_i^1}{k+1} \right\|^2 \right\},$$

then we get by triangle inequality:

$$
\begin{aligned}
r^{\tan}(\pi^t) &\leq \left\| -V(\pi^t) + V(\pi^{t-1}) - \frac{\mu}{k(t-1)+1}(\sigma^{k(t-1)} - \sigma^1) \right. \\
&\quad \left. - \mu(\pi^{\mu,k(t-1)} - \pi^{\mu,k(t-1)} + \pi^{t-1} - \sigma^{k(t-1)}) - \frac{1}{\eta}(\pi^t - \pi^{t-1}) \right\| \\
&\leq \left\| -V(\pi^t) + V(\pi^{t-1}) \right\| + \frac{\mu}{k(t-1)+1} \left\| \sigma^{k(t-1)} - \sigma^1 \right\| \\
&\quad + \mu \left\| \pi^{\mu,k(t-1)} - \sigma^{k(t-1)} \right\| + \mu \left\| \pi^{\mu,k(t-1)} - \pi^{t-1} \right\| + \frac{1}{\eta} \left\| \pi^t - \pi^{t-1} \right\| \\
&\leq \frac{1+\eta L}{\eta} \left\| \pi^t - \pi^{t-1} \right\| + \frac{\mu D}{k(t-1)+1} \\
&\quad + \mu \left\| \pi^{\mu,k(t-1)} - \sigma^{k(t-1)} \right\| + \mu \left\| \pi^{\mu,k(t-1)} - \pi^{t-1} \right\| \\
&\leq \frac{1+\eta L}{\eta} \left\| \pi^{\mu,k(t-1)} - \pi^t \right\| + \frac{\mu D}{k(t-1)+1} + \mu \left\| \pi^{\mu,k(t-1)} - \sigma^{k(t-1)} \right\| \\
&\quad + \left( \mu + \frac{1+\eta L}{\eta} \right) \left\| \pi^{\mu,k(t-1)} - \pi^{t-1} \right\|. \tag{10}
\end{aligned}
$$

In terms of upper bound on $\left\| \pi^{\mu,k(t-1)} - \pi^t \right\|$ and $\left\| \pi^{\mu,k(t-1)} - \pi^{t-1} \right\|$, we introduce the following lemma:

**Lemma B.1.** *If we use the constant learning rate $\eta_t = \eta \in (0, \frac{\mu}{(L+\mu)^2})$, we have for any $t \geq 1$:*

$$
\left\| \pi^{\mu,k(t)} - \pi^t \right\|^2 \leq \left( \frac{1}{1+\eta\mu} \right)^{t-(k(t)-1)T_\sigma - 1} \left\| \pi^{\mu,k(t)} - \sigma^{k(t)} \right\|^2,
$$

$$
\left\| \pi^{\mu,k(t)} - \pi^{t+1} \right\|^2 \leq \left( \frac{1}{1+\eta\mu} \right)^{t-(k(t)-1)T_\sigma} \left\| \pi^{\mu,k(t)} - \sigma^{k(t)} \right\|^2.
$$

Combining Eq. (10) and Lemma B.1, we have for any $t \geq 2$:

$$
r^{\tan}(\pi^t) \leq 2 \left( \mu + \frac{1+\eta L}{\eta} \right) \left\| \pi^{\mu,k(t-1)} - \sigma^{k(t-1)} \right\| + \frac{\mu D}{k(t-1)+1}. \tag{11}
$$

Next, we derive the following upper bound on $\left\| \pi^{\mu,k(t-1)} - \sigma^{k(t-1)} \right\|$:

**Lemma B.2.** *If we set $\eta_t = \eta \in (0, \frac{\mu}{(L+\mu)^2})$ and $T_\sigma \geq \max(1, \frac{6\ln 3(T+1)}{\ln(1+\eta\mu)})$, we have for any $t \geq 1$:*

$$
\left\| \pi^{\mu,k(t)} - \sigma^{k(t)} \right\| \leq \frac{8D}{k(t)+1}.
$$

By combining Eq. (11) and Lemma B.2, we get:

$$
\begin{aligned}
r^{\tan}(\pi^t) &\leq \frac{16D}{k(t-1)+1} \left( \mu + \frac{1+\eta L}{\eta} \right) + \frac{\mu D}{k(t-1)+1} \\
&\leq \frac{17D}{k(t-1)+1} \left( \mu + \frac{1+\eta L}{\eta} \right).
\end{aligned}
$$

Therefore, since $k(t) = \lfloor \frac{t-1}{T_\sigma} \rfloor + 1$, it holds that:

$$
r^{\tan}(\pi^t) \leq \frac{17DT_\sigma}{t + T_\sigma - 2} \left( \mu + \frac{1+\eta L}{\eta} \right).
$$

Finally, taking $T_\sigma = c \cdot \max(1, \frac{6\ln 3(T+1)}{\ln(1+\eta\mu)})$, we have for any $t \geq 2$:

$$
r^{\tan}(\pi^t) \leq \frac{17cD \left( \frac{6\ln 3(T+1)}{\ln(1+\eta\mu)} + 1 \right)}{t-1} \left( \mu + \frac{1+\eta L}{\eta} \right).
$$

$\square$

## B.2 PROOF OF LEMMA B.1

*Proof of Lemma B.1.* First, we have for any three vectors $a, b, c$:

$$\frac{1}{2}\left\|a-b\right\|^2 - \frac{1}{2}\left\|a-c\right\|^2 + \frac{1}{2}\left\|b-c\right\|^2 = \langle c-b, a-b\rangle.$$

Thus, we have for any $t \geq 1$:

$$\frac{1}{2}\left\|\pi^{\mu,k(t)}-\pi^{t+1}\right\|^2 - \frac{1}{2}\left\|\pi^{\mu,k(t)}-\pi^t\right\|^2 + \frac{1}{2}\left\|\pi^{t+1}-\pi^t\right\|^2 = \left\langle \pi^t-\pi^{t+1}, \pi^{\mu,k(t)}-\pi^{t+1}\right\rangle. \tag{12}$$

Here, let us define $\hat{\sigma}^{k(t)} = \frac{k(t)\sigma^{k(t)}+\sigma^1}{k(t)+1}$. Then, from the first-order optimality condition for $\pi^{t+1}$, we have for any $t \geq 1$:

$$\left\langle \eta\left(V(\pi^t) - \mu\left(\pi^t - \hat{\sigma}^{k(t)}\right)\right) - \pi^{t+1} + \pi^t, \pi^{t+1} - \pi^{\mu,k(t)}\right\rangle \geq 0. \tag{13}$$

Similarly, from the first-order optimality condition for $\pi^{\mu,k(t)}$, we get:

$$\left\langle V(\pi^{\mu,k(t)}) - \mu\left(\pi^{\mu,k(t)} - \hat{\sigma}^{k(t)}\right), \pi^{\mu,k(t)} - \pi^{t+1}\right\rangle \geq 0. \tag{14}$$

Combining Eq. (12), Eq. (13), and Eq. (14) yields:

$$\begin{aligned}
&\frac{1}{2}\left\|\pi^{\mu,k(t)}-\pi^{t+1}\right\|^2 - \frac{1}{2}\left\|\pi^{\mu,k(t)}-\pi^t\right\|^2 + \frac{1}{2}\left\|\pi^{t+1}-\pi^t\right\|^2 \\
&\leq \eta\left\langle V(\pi^t) - \mu\left(\pi^t - \hat{\sigma}^{k(t)}\right), \pi^{t+1} - \pi^{\mu,k(t)}\right\rangle \\
&= \eta\left\langle V(\pi^{t+1}) - \mu\left(\pi^{t+1} - \hat{\sigma}^{k(t)}\right), \pi^{t+1} - \pi^{\mu,k(t)}\right\rangle \\
&\quad + \eta\left\langle V(\pi^t) - V(\pi^{t+1}) - \mu\left(\pi^t - \pi^{t+1}\right), \pi^{t+1} - \pi^{\mu,k(t)}\right\rangle \\
&\leq \eta\left\langle V(\pi^{\mu,k(t)}) - \mu\left(\pi^{t+1} - \hat{\sigma}^{k(t)}\right), \pi^{t+1} - \pi^{\mu,k(t)}\right\rangle \\
&\quad + \eta\left\langle V(\pi^t) - V(\pi^{t+1}) - \mu\left(\pi^t - \pi^{t+1}\right), \pi^{t+1} - \pi^{\mu,k(t)}\right\rangle \\
&= \eta\left\langle V(\pi^{\mu,k(t)}) - \mu\left(\pi^{\mu,k(t)} - \hat{\sigma}^{k(t)}\right), \pi^{t+1} - \pi^{\mu,k(t)}\right\rangle - \eta\mu\left\|\pi^{t+1} - \pi^{\mu,k(t)}\right\|^2 \\
&\quad + \eta\left\langle V(\pi^t) - V(\pi^{t+1}) - \mu\left(\pi^t - \pi^{t+1}\right), \pi^{t+1} - \pi^{\mu,k(t)}\right\rangle \\
&\leq -\eta\mu\left\|\pi^{t+1} - \pi^{\mu,k(t)}\right\|^2 + \eta\left\langle V(\pi^t) - V(\pi^{t+1}) - \mu\left(\pi^t - \pi^{t+1}\right), \pi^{t+1} - \pi^{\mu,k(t)}\right\rangle, \tag{15}
\end{aligned}$$

where the second inequality follows from Eq. (1). From Cauchy-Schwarz inequality and Young's inequality, the second term in the right-hand side of this inequality can be bounded by:

$$\begin{aligned}
&\eta\left\langle V(\pi^t) - V(\pi^{t+1}) - \mu\left(\pi^t - \pi^{t+1}\right), \pi^{t+1} - \pi^{\mu,k(t)}\right\rangle \\
&= \eta\left\langle V(\pi^t) - V(\pi^{t+1}), \pi^{t+1} - \pi^{\mu,k(t)}\right\rangle - \eta\mu\left\langle \pi^t - \pi^{t+1}, \pi^{t+1} - \pi^{\mu,k(t)}\right\rangle \\
&\leq \eta\left(\left\|V(\pi^t) - V(\pi^{t+1})\right\| + \mu\left\|\pi^t - \pi^{t+1}\right\|\right) \cdot \left\|\pi^{t+1} - \pi^{\mu,k(t)}\right\| \\
&\leq \eta(L+\mu)\left\|\pi^t - \pi^{t+1}\right\| \cdot \left\|\pi^{t+1} - \pi^{\mu,k(t)}\right\| \\
&\leq \frac{1}{2}\left\|\pi^t - \pi^{t+1}\right\|^2 + \frac{\eta^2(L+\mu)^2}{2}\left\|\pi^{t+1} - \pi^{\mu,k(t)}\right\|^2 \\
&\leq \frac{1}{2}\left\|\pi^t - \pi^{t+1}\right\|^2 + \frac{\eta\mu}{2}\left\|\pi^{t+1} - \pi^{\mu,k(t)}\right\|^2, \tag{16}
\end{aligned}$$

where the second inequality follow from Eq. (2), and the last inequality follows from the assumption that $\eta \leq \frac{\mu}{(L+\mu)^2}$. By combining Eq. (15) and Eq. (16), we get:

$$
\frac{1}{2} \left\| \pi^{\mu,k(t)} - \pi^{t+1} \right\|^2 - \frac{1}{2} \left\| \pi^{\mu,k(t)} - \pi^t \right\|^2 + \frac{1}{2} \left\| \pi^{t+1} - \pi^t \right\|^2
$$
$$
\leq -\frac{\eta\mu}{2} \left\| \pi^{t+1} - \pi^{\mu,k(t)} \right\|^2 + \frac{1}{2} \left\| \pi^t - \pi^{t+1} \right\|^2 .
$$

Thus,

$$
\frac{1+\eta\mu}{2} \left\| \pi^{\mu,k(t)} - \pi^{t+1} \right\|^2 \leq \frac{1}{2} \left\| \pi^{\mu,k(t)} - \pi^t \right\|^2 .
$$

Therefore, we have for any $t \geq 1$:

$$
\left\| \pi^{\mu,k(t)} - \pi^{t+1} \right\|^2 \leq \frac{1}{1+\eta\mu} \left\| \pi^{\mu,k(t)} - \pi^t \right\|^2 .
$$

Furthermore, since $k(s) = k(t)$ for $s \in [(k(t)-1)T_\sigma + 1, t]$, we have for such $s$ that:

$$
\left\| \pi^{\mu,k(t)} - \pi^{s+1} \right\|^2 \leq \frac{1}{1+\eta\mu} \left\| \pi^{\mu,k(t)} - \pi^s \right\|^2 .
$$

Therefore, by applying this inequality from $t, t-1, \cdots, (k(t)-1)T_\sigma + 1$, we get for any $t \geq 1$:

$$
\left\| \pi^{\mu,k(t)} - \pi^{t+1} \right\|^2 \leq \left( \frac{1}{1+\eta\mu} \right)^{t-(k(t)-1)T_\sigma} \left\| \pi^{\mu,k(t)} - \pi^{(k(t)-1)T_\sigma+1} \right\|^2
$$
$$
= \left( \frac{1}{1+\eta\mu} \right)^{t-(k(t)-1)T_\sigma} \left\| \pi^{\mu,k(t)} - \sigma^{k(t)} \right\|^2 . \tag{17}
$$

Here, since $k(t) = k(t+1)$ when $t$ satisfies that $t \neq T_\sigma \left\lfloor \frac{t}{T_\sigma} \right\rfloor$, we have for such $t$ that:

$$
\left\| \pi^{\mu,k(t+1)} - \pi^{t+1} \right\|^2 \leq \left( \frac{1}{1+\eta\mu} \right)^{t-(k(t+1)-1)T_\sigma} \left\| \pi^{\mu,k(t+1)} - \sigma^{k(t+1)} \right\|^2 . \tag{18}
$$

On the other hand, when $t$ satisfies that $t = T_\sigma \left\lfloor \frac{t}{T_\sigma} \right\rfloor$:

$$
k(t+1) = \left\lceil \frac{T_\sigma \left\lfloor \frac{t}{T_\sigma} \right\rfloor + 1 - 1}{T_\sigma} \right\rceil + 1 = \left\lfloor \frac{t}{T_\sigma} \right\rfloor + 1
$$
$$
\Rightarrow (k(t+1)-1)T_\sigma = T_\sigma \left\lfloor \frac{t}{T_\sigma} \right\rfloor = t
$$
$$
\Rightarrow \pi^{t+1} = \pi^{(k(t+1)-1)T_\sigma+1} = \sigma^{k(t+1)} .
$$

Therefore, we have for any $t \geq 1$ such that $t = T_\sigma \left\lfloor \frac{t}{T_\sigma} \right\rfloor$:

$$
\left\| \pi^{\mu,k(t+1)} - \pi^{t+1} \right\|^2 = \left\| \pi^{\mu,k(t+1)} - \sigma^{k(t+1)} \right\|^2
$$
$$
= \left( \frac{1}{1+\eta\mu} \right)^{t-(k(t+1)-1)T_\sigma} \left\| \pi^{\mu,k(t+1)} - \sigma^{k(t+1)} \right\|^2 . \tag{19}
$$

By combining Eq. (17), Eq. (18), and Eq. (19), we have for any $t \geq 1$:

$$
\left\| \pi^{\mu,k(t)} - \pi^{t+1} \right\|^2 \leq \left( \frac{1}{1+\eta\mu} \right)^{t-(k(t)-1)T_\sigma} \left\| \pi^{\mu,k(t)} - \sigma^{k(t)} \right\|^2 ,
$$
$$
\left\| \pi^{\mu,k(t+1)} - \pi^{t+1} \right\|^2 \leq \left( \frac{1}{1+\eta\mu} \right)^{t-(k(t+1)-1)T_\sigma} \left\| \pi^{\mu,k(t+1)} - \sigma^{k(t+1)} \right\|^2 .
$$

$\square$

### B.3 PROOF OF LEMMA B.2

*Proof of Lemma B.2.* First, we have for any Nash equilibrium $\pi^* \in \Pi^*$ and $t \geq 1$ such that $k(t) \geq 1$:

$$
\begin{aligned}
&\frac{(k(t)+1)(k(t)+2)}{2} \left\| \pi^{\mu,k(t)} - \hat{\sigma}^{k(t)} \right\|^2 + (k(t)+1)(k(t)+2) \left\langle \hat{\sigma}^{k(t)+1} - \pi^{\mu,k(t)}, \pi^{\mu,k(t)} - \hat{\sigma}^{k(t)} \right\rangle \\
&= \frac{(k(t)+1)(k(t)+2)}{2} \left\| \pi^{\mu,k(t)} - \hat{\sigma}^{k(t)} \right\|^2 \\
&\quad + (k(t)+1) \left\langle (k(t)+1)\sigma^{k(t)+1} + \sigma^1 - (k(t)+2)\pi^{\mu,k(t)}, \pi^{\mu,k(t)} - \hat{\sigma}^{k(t)} \right\rangle \\
&= \frac{(k(t)+1)(k(t)+2)}{2} \left\| \pi^{\mu,k(t)} - \hat{\sigma}^{k(t)} \right\|^2 + (k(t)+1) \left\langle \sigma^1 - \sigma^{k(t)+1}, \pi^{\mu,k(t)} - \hat{\sigma}^{k(t)} \right\rangle \\
&\quad + (k(t)+1)(k(t)+2) \left\langle \sigma^{k(t)+1} - \pi^{\mu,k(t)}, \pi^{\mu,k(t)} - \hat{\sigma}^{k(t)} \right\rangle \\
&= \frac{(k(t)+1)(k(t)+2)}{2} \left\| \pi^{\mu,k(t)} - \hat{\sigma}^{k(t)} \right\|^2 + (k(t)+1) \left\langle \sigma^1 - \pi^{\mu,k(t)}, \pi^{\mu,k(t)} - \hat{\sigma}^{k(t)} \right\rangle \\
&\quad + (k(t)+1)^2 \left\langle \sigma^{k(t)+1} - \pi^{\mu,k(t)}, \pi^{\mu,k(t)} - \hat{\sigma}^{k(t)} \right\rangle \\
&= \frac{(k(t)+1)(k(t)+2)}{2} \left\| \pi^{\mu,k(t)} - \hat{\sigma}^{k(t)} \right\|^2 + (k(t)+1) \left\langle \sigma^1 - \pi^*, \pi^{\mu,k(t)} - \hat{\sigma}^{k(t)} \right\rangle \\
&\quad + (k(t)+1) \left\langle \pi^* - \pi^{\mu,k(t)}, \pi^{\mu,k(t)} - \hat{\sigma}^{k(t)} \right\rangle + (k(t)+1)^2 \left\langle \sigma^{k(t)+1} - \pi^{\mu,k(t)}, \pi^{\mu,k(t)} - \hat{\sigma}^{k(t)} \right\rangle.
\end{aligned}
$$

Here, the first-order optimality condition for $\pi^{\mu,k(t)}$:

$$
\begin{aligned}
&\left\langle V(\pi^{\mu,k(t)}) - \mu\left(\pi^{\mu,k(t)} - \hat{\sigma}^{k(t)}\right), \pi^{\mu,k(t)} - \pi^* \right\rangle \geq 0 \\
&\Rightarrow \left\langle \pi^{\mu,k(t)} - \hat{\sigma}^{k(t)}, \pi^* - \pi^{\mu,k(t)} \right\rangle \geq \frac{1}{\mu} \left\langle V(\pi^{\mu,k(t)}), \pi^* - \pi^{\mu,k(t)} \right\rangle \geq \frac{1}{\mu} \left\langle V(\pi^*), \pi^* - \pi^{\mu,k(t)} \right\rangle \geq 0,
\end{aligned}
$$

where we use Eq. (1) and the fact that $\pi^*$ is a Nash equilibrium. Combining these inequalities yields:

$$
\begin{aligned}
&\frac{(k(t)+1)(k(t)+2)}{2} \left\| \pi^{\mu,k(t)} - \hat{\sigma}^{k(t)} \right\|^2 + (k(t)+1)(k(t)+2) \left\langle \hat{\sigma}^{k(t)+1} - \pi^{\mu,k(t)}, \pi^{\mu,k(t)} - \hat{\sigma}^{k(t)} \right\rangle \\
&\geq \frac{(k(t)+1)(k(t)+2)}{2} \left\| \pi^{\mu,k(t)} - \hat{\sigma}^{k(t)} \right\|^2 + (k(t)+1) \left\langle \sigma^1 - \pi^*, \pi^{\mu,k(t)} - \hat{\sigma}^{k(t)} \right\rangle \\
&\quad + (k(t)+1)^2 \left\langle \sigma^{k(t)+1} - \pi^{\mu,k(t)}, \pi^{\mu,k(t)} - \hat{\sigma}^{k(t)} \right\rangle.
\end{aligned}
$$

From Young's inequality, we have for any $\rho_1, \rho_2 > 0$:

$$
\begin{aligned}
&\frac{(k(t)+1)(k(t)+2)}{2} \left\| \pi^{\mu,k(t)} - \hat{\sigma}^{k(t)} \right\|^2 + (k(t)+1)(k(t)+2) \left\langle \hat{\sigma}^{k(t)+1} - \pi^{\mu,k(t)}, \pi^{\mu,k(t)} - \hat{\sigma}^{k(t)} \right\rangle \\
&\geq \frac{(k(t)+1)(k(t)+2)}{2} \left\| \pi^{\mu,k(t)} - \hat{\sigma}^{k(t)} \right\|^2 - \frac{\rho_1(k(t)+1)}{2} \left\| \sigma^1 - \pi^* \right\|^2 - \frac{(k(t)+1)}{2\rho_1} \left\| \pi^{\mu,k(t)} - \hat{\sigma}^{k(t)} \right\|^2 \\
&\quad - \frac{\rho_2(k(t)+1)^2}{2} \left\| \sigma^{k(t)+1} - \pi^{\mu,k(t)} \right\|^2 - \frac{(k(t)+1)^2}{2\rho_2} \left\| \pi^{\mu,k(t)} - \hat{\sigma}^{k(t)} \right\|^2 \\
&= \left( \frac{(k(t)+1)(k(t)+2)}{2} - \frac{k(t)+1}{2\rho_1} - \frac{(k(t)+1)^2}{2\rho_2} \right) \left\| \pi^{\mu,k(t)} - \hat{\sigma}^{k(t)} \right\|^2 \\
&\quad - \frac{\rho_1(k(t)+1)}{2} \left\| \sigma^1 - \pi^* \right\|^2 - \frac{\rho_2(k(t)+1)^2}{2} \left\| \sigma^{k(t)+1} - \pi^{\mu,k(t)} \right\|^2.
\end{aligned}
$$

By setting $\rho_1 = \frac{4}{k(t)+2}, \rho_2 = \frac{4(k(t)+1)}{k(t)+2}$, we obtain:

$$\frac{(k(t)+1)(k(t)+2)}{2} \left\| \pi^{\mu,k(t)} - \hat{\sigma}^{k(t)} \right\|^2 + (k(t)+1)(k(t)+2) \left\langle \hat{\sigma}^{k(t)+1} - \pi^{\mu,k(t)}, \pi^{\mu,k(t)} - \hat{\sigma}^{k(t)} \right\rangle$$

$$\geq \frac{(k(t)+1)(k(t)+2)}{4} \left\| \pi^{\mu,k(t)} - \hat{\sigma}^{k(t)} \right\|^2 - \frac{2(k(t)+1)}{k(t)+2} \left\| \sigma^1 - \pi^* \right\|^2$$

$$- \frac{2(k(t)+1)^3}{k(t)+2} \left\| \sigma^{k(t)+1} - \pi^{\mu,k(t)} \right\|^2$$

$$\geq \frac{(k(t)+1)(k(t)+2)}{4} \left\| \pi^{\mu,k(t)} - \hat{\sigma}^{k(t)} \right\|^2 - 2 \left\| \sigma^1 - \pi^* \right\|^2 - 2(k(t)+1)^2 \left\| \sigma^{k(t)+1} - \pi^{\mu,k(t)} \right\|^2.$$

$$(20)$$

Here, we introduce the following lemma:

**Lemma B.3.** *For any $t \geq 1$ such that $k(t) \geq 2$, we have:*

$$\frac{(k(t)+1)(k(t)+2)}{2} \left\| \pi^{\mu,k(t)} - \hat{\sigma}^{k(t)} \right\|^2 + (k(t)+1)(k(t)+2) \left\langle \hat{\sigma}^{k(t)+1} - \pi^{\mu,k(t)}, \pi^{\mu,k(t)} - \hat{\sigma}^{k(t)} \right\rangle$$

$$\leq \frac{k(t)(k(t)+1)}{2} \left\| \pi^{\mu,k(t)-1} - \hat{\sigma}^{k(t)-1} \right\|^2 + k(t)(k(t)+1) \left\langle \hat{\sigma}^{k(t)} - \pi^{\mu,k(t)-1}, \pi^{\mu,k(t)-1} - \hat{\sigma}^{k(t)-1} \right\rangle$$

$$+ (k(t)+1) \left\langle (k(t)+1)(\pi^{\mu,k(t)} - \sigma^{k(t)+1}) + k(t)(\sigma^{k(t)} - \pi^{\mu,k(t)-1}), \hat{\sigma}^{k(t)} - \pi^{\mu,k(t)} \right\rangle.$$

By combining Eq. (20) and Lemma B.3, we get:

$$\frac{(k(t)+1)(k(t)+2)}{4} \left\| \pi^{\mu,k(t)} - \hat{\sigma}^{k(t)} \right\|^2$$

$$\leq \frac{(k(t)+1)(k(t)+2)}{2} \left\| \pi^{\mu,k(t)} - \hat{\sigma}^{k(t)} \right\|^2 + (k(t)+1)(k(t)+2) \left\langle \hat{\sigma}^{k(t)+1} - \pi^{\mu,k(t)}, \pi^{\mu,k(t)} - \hat{\sigma}^{k(t)} \right\rangle$$

$$+ 2 \left\| \sigma^1 - \pi^* \right\|^2 + 2(k(t)+1)^2 \left\| \sigma^{k(t)+1} - \pi^{\mu,k(t)} \right\|^2$$

$$\leq 3 \left\| \pi^{\mu,1} - \hat{\sigma}^1 \right\|^2 + 6 \left\langle \hat{\sigma}^2 - \pi^{\mu,1}, \pi^{\mu,1} - \hat{\sigma}^1 \right\rangle + 2 \left\| \sigma^1 - \pi^* \right\|^2 + 2(k(t)+1)^2 \left\| \sigma^{k(t)+1} - \pi^{\mu,k(t)} \right\|^2$$

$$+ \sum_{l=2}^{k(t)} (l+1) \left\langle (l+1)(\pi^{\mu,l} - \sigma^{l+1}) + l(\sigma^l - \pi^{\mu,l-1}), \hat{\sigma}^l - \pi^{\mu,l} \right\rangle$$

$$= 3 \left\| \pi^{\mu,1} - \sigma^1 \right\|^2 + 2 \left\langle 2\sigma^2 + \sigma^1 - 3\pi^{\mu,1}, \pi^{\mu,1} - \sigma^1 \right\rangle + 2 \left\| \sigma^1 - \pi^* \right\|^2$$

$$+ 2(k(t)+1)^2 \left\| \sigma^{k(t)+1} - \pi^{\mu,k(t)} \right\|^2 + \sum_{l=2}^{k(t)} (l+1) \left\langle (l+1)(\pi^{\mu,l} - \sigma^{l+1}) + l(\sigma^l - \pi^{\mu,l-1}), \hat{\sigma}^l - \pi^{\mu,l} \right\rangle$$

$$= 3 \left\| \pi^{\mu,1} - \sigma^1 \right\|^2 + 2 \left\langle \sigma^1 - \pi^{\mu,1}, \pi^{\mu,1} - \sigma^1 \right\rangle + 4 \left\langle \sigma^2 - \pi^{\mu,1}, \pi^{\mu,1} - \sigma^1 \right\rangle$$

$$+ 2 \left\| \sigma^1 - \pi^* \right\|^2 + 2(k(t)+1)^2 \left\| \sigma^{k(t)+1} - \pi^{\mu,k(t)} \right\|^2$$

$$+ \sum_{l=2}^{k(t)} (l+1) \left\langle (l+1)(\pi^{\mu,l} - \sigma^{l+1}) + l(\sigma^l - \pi^{\mu,l-1}), \hat{\sigma}^l - \pi^{\mu,l} \right\rangle$$

$$= \left\| \pi^{\mu,1} - \sigma^1 \right\|^2 + 4 \left\langle \sigma^2 - \pi^{\mu,1}, \pi^{\mu,1} - \sigma^1 \right\rangle + 2 \left\| \sigma^1 - \pi^* \right\|^2 + 2(k(t)+1)^2 \left\| \sigma^{k(t)+1} - \pi^{\mu,k(t)} \right\|^2$$

$$+ \sum_{l=2}^{k(t)} (l+1) \left\langle (l+1)(\pi^{\mu,l} - \sigma^{l+1}) + l(\sigma^l - \pi^{\mu,l-1}), \hat{\sigma}^l - \pi^{\mu,l} \right\rangle$$

$$= \left\| \pi^{\mu,1} - \sigma^1 \right\|^2 + 2 \left\| \sigma^1 - \pi^* \right\|^2 + 2(k(t)+1)^2 \left\| \sigma^{k(t)+1} - \pi^{\mu,k(t)} \right\|^2$$

$$+ \sum_{l=1}^{k(t)} (l+1)^2 \left\langle \pi^{\mu,l} - \sigma^{l+1}, \hat{\sigma}^l - \pi^{\mu,l} \right\rangle + \sum_{l=2}^{k(t)} l(l+1) \left\langle \sigma^l - \pi^{\mu,l-1}, \hat{\sigma}^l - \pi^{\mu,l} \right\rangle$$

$$\leq 3D^2 + 2(k(t)+1)^2 \left\|\sigma^{k(t)+1} - \pi^{\mu,k(t)}\right\|^2 + 2D(k(t)+1)^2 \sum_{l=1}^{k(t)} \left\|\pi^{\mu,l} - \sigma^{l+1}\right\|.$$

Therefore, we have for any $t \geq 1$ such that $k(t) \geq 2$:

$$\left\|\pi^{\mu,k(t)} - \hat{\sigma}^{k(t)}\right\|^2 \leq \frac{12D^2}{(k(t)+1)^2} + 8\left\|\sigma^{k(t)+1} - \pi^{\mu,k(t)}\right\|^2 + 8D \sum_{l=1}^{k(t)} \left\|\pi^{\mu,l} - \sigma^{l+1}\right\|.$$

By the definition of $\hat{\sigma}^{k(t)}$,

$$\left\|\pi^{\mu,k(t)} - \sigma^{k(t)}\right\|^2 + \frac{\left\|\sigma^{k(t)} - \sigma^1\right\|^2}{(k(t)+1)^2} + \frac{2}{k(t)+1}\left\langle \pi^{\mu,k(t)} - \sigma^{k(t)}, \sigma^{k(t)} - \sigma^1 \right\rangle$$

$$\leq \frac{12D^2}{(k(t)+1)^2} + 8\left\|\sigma^{k(t)+1} - \pi^{\mu,k(t)}\right\|^2 + 8D \sum_{l=1}^{k(t)} \left\|\pi^{\mu,l} - \sigma^{l+1}\right\|.$$

Therefore, from Cauchy-Schwarz inequality, we have:

$$\left\|\pi^{\mu,k(t)} - \sigma^{k(t)}\right\|^2$$

$$\leq \frac{2}{k(t)+1}\left\langle \pi^{\mu,k(t)} - \sigma^{k(t)}, \sigma^1 - \sigma^{k(t)} \right\rangle + \frac{12D^2}{(k(t)+1)^2}$$

$$+ 8\left\|\sigma^{k(t)+1} - \pi^{\mu,k(t)}\right\|^2 + 8D \sum_{l=1}^{k(t)} \left\|\pi^{\mu,l} - \sigma^{l+1}\right\|$$

$$\leq \frac{2D}{k(t)+1}\left\|\pi^{\mu,k(t)} - \sigma^{k(t)}\right\| + \frac{12D^2}{(k(t)+1)^2} + 8\left\|\sigma^{k(t)+1} - \pi^{\mu,k(t)}\right\|^2 + 8D \sum_{l=1}^{k(t)} \left\|\pi^{\mu,l} - \sigma^{l+1}\right\|. \tag{21}$$

Furthermore, from Lemma B.1, we have for any $k \geq 1$:

$$\left\|\pi^{\mu,k} - \sigma^{k+1}\right\|^2 \leq \left(\frac{1}{1+\eta\mu}\right)^{T_\sigma} \left\|\pi^{\mu,k} - \sigma^k\right\|^2. \tag{22}$$

Combining Eq. (21) nad Eq. (22), we have for any $t \geq 1$ such that $k(t) \geq 2$:

$$\left\|\pi^{\mu,k(t)} - \sigma^{k(t)}\right\|^2 \leq \frac{2D}{k(t)+1}\left\|\pi^{\mu,k(t)} - \sigma^{k(t)}\right\| + \frac{12D^2}{(k(t)+1)^2}$$

$$+ 8\left(\frac{1}{1+\eta\mu}\right)^{T_\sigma} \left\|\pi^{\mu,k(t)} - \sigma^{k(t)}\right\|^2 + 8D^2 k(t)\left(\frac{1}{1+\eta\mu}\right)^{\frac{T_\sigma}{2}}.$$

Therefore, since $T_\sigma \geq \max(1, \frac{6\ln 3(T+1)}{\ln(1+\eta\mu)}) \Rightarrow \left(\frac{1}{1+\eta\mu}\right)^{T_\sigma} \leq \frac{(k(t)+1)^3}{(1+\eta\mu)^{T_\sigma}} \leq \frac{1}{16}$, we have for $k(t) \geq 2$:

$$\frac{1}{2}\left(\left\|\pi^{\mu,k(t)} - \sigma^{k(t)}\right\| - \frac{2D}{k(t)+1}\right)^2 \leq \frac{2D^2}{(k(t)+1)^2} + \frac{12D^2}{(k(t)+1)^2} + \frac{D^2}{2(k(t)+1)^2} \leq \frac{16D^2}{(k(t)+1)^2},$$

and then:

$$\left\|\pi^{\mu,k(t)} - \sigma^{k(t)}\right\| \leq \frac{2D}{k(t)+1} + \frac{4\sqrt{2}D}{k(t)+1} \leq \frac{8D}{k(t)+1}.$$

On the other hand, for $k(t) = 1$, we have:

$$\left\|\pi^{\mu,1} - \sigma^1\right\| \leq D \leq \frac{8D}{1+1}.$$

In summary, for any $t \geq 1$, we have:

$$\left\| \pi^{\mu, k(t)} - \sigma^{k(t)} \right\| \leq \frac{8D}{k(t) + 1}.$$

$\square$

### B.4 PROOF OF LEMMA B.3

*Proof of Lemma B.3.* From the first-order optimality condition for $\pi^{\mu, k(t)}$, we have:

$$\left\langle V(\pi^{\mu, k(t)}) - \mu(\pi^{\mu, k(t)} - \hat{\sigma}^{k(t)}), \pi^{\mu, k(t)} - \pi^{\mu, k(t)-1} \right\rangle \geq 0.$$

Similarly, from the first-order optimality condition for $\pi^{\mu, k(t)-1}$, we have:

$$\left\langle V(\pi^{\mu, k(t)-1}) - \mu(\pi^{\mu, k(t)-1} - \hat{\sigma}^{k(t)-1}), \pi^{\mu, k(t)-1} - \pi^{\mu, k(t)} \right\rangle \geq 0.$$

Summing up these inequalities, we get for any $t \geq 1$ such that $k(t) \geq 2$:

$$0 \leq \left\langle V(\pi^{\mu, k(t)}) - V(\pi^{\mu, k(t)-1}), \pi^{\mu, k(t)} - \pi^{\mu, k(t)-1} \right\rangle - \mu \left\langle \pi^{\mu, k(t)} - \hat{\sigma}^{k(t)}, \pi^{\mu, k(t)} - \pi^{\mu, k(t)-1} \right\rangle$$

$$+ \mu \left\langle \hat{\sigma}^{k(t)-1} - \pi^{\mu, k(t)-1}, \pi^{\mu, k(t)-1} - \pi^{\mu, k(t)} \right\rangle$$

$$\leq -\mu \left\langle \pi^{\mu, k(t)} - \hat{\sigma}^{k(t)}, \pi^{\mu, k(t)} - \pi^{\mu, k(t)-1} \right\rangle + \mu \left\langle \hat{\sigma}^{k(t)-1} - \pi^{\mu, k(t)-1}, \pi^{\mu, k(t)-1} - \pi^{\mu, k(t)} \right\rangle$$

$$= -\mu \left\langle \pi^{\mu, k(t)} - \sigma^{k(t)} + \sigma^{k(t)} - \hat{\sigma}^{k(t)}, \pi^{\mu, k(t)} - \sigma^{k(t)} + \sigma^{k(t)} - \pi^{\mu, k(t)-1} \right\rangle$$

$$+ \mu \left\langle \hat{\sigma}^{k(t)-1} - \pi^{\mu, k(t)-1}, \pi^{\mu, k(t)-1} - \pi^{\mu, k(t)} \right\rangle$$

$$= -\mu \left\| \pi^{\mu, k(t)} - \sigma^{k(t)} \right\|^2 - \mu \left\langle \pi^{\mu, k(t)} - \sigma^{k(t)}, \sigma^{k(t)} - \pi^{\mu, k(t)-1} \right\rangle$$

$$- \mu \left\langle \sigma^{k(t)} - \hat{\sigma}^{k(t)}, \pi^{\mu, k(t)} - \pi^{\mu, k(t)-1} \right\rangle + \mu \left\langle \hat{\sigma}^{k(t)-1} - \pi^{\mu, k(t)-1}, \pi^{\mu, k(t)-1} - \pi^{\mu, k(t)} \right\rangle$$

$$= -\mu \left\| \pi^{\mu, k(t)} - \sigma^{k(t)} \right\|^2 - \mu \left\langle \pi^{\mu, k(t)} - \sigma^{k(t)}, \sigma^{k(t)} - \pi^{\mu, k(t)-1} \right\rangle$$

$$+ \mu \left\langle \pi^{\mu, k(t)} - \pi^{\mu, k(t)-1}, \pi^{\mu, k(t)-1} - \sigma^{k(t)} \right\rangle + \mu \left\langle \hat{\sigma}^{k(t)-1} - \hat{\sigma}^{k(t)}, \pi^{\mu, k(t)-1} - \pi^{\mu, k(t)} \right\rangle.$$

Here, for any vectors $a, b, c$, it holds that:

$$\langle a - b, b - c \rangle = \frac{1}{2} \|a - c\|^2 - \frac{1}{2} \|b - c\|^2 - \frac{1}{2} \|a - b\|^2,$$

$$\langle a - b, c - d \rangle = \frac{1}{2} \|a - b\|^2 + \frac{1}{2} \|c - d\|^2 - \frac{1}{2} \|d - c + a - b\|^2.$$

Thus, we have:

$$0 \leq -\mu \left\| \pi^{\mu, k(t)} - \sigma^{k(t)} \right\|^2 - \frac{\mu}{2} \left\| \pi^{\mu, k(t)} - \pi^{\mu, k(t)-1} \right\|^2$$

$$+ \frac{\mu}{2} \left\| \pi^{\mu, k(t)-1} - \sigma^{k(t)} \right\|^2 + \frac{\mu}{2} \left\| \pi^{\mu, k(t)} - \sigma^{k(t)} \right\|^2$$

$$+ \frac{\mu}{2} \left\| \pi^{\mu, k(t)} - \sigma^{k(t)} \right\|^2 - \frac{\mu}{2} \left\| \pi^{\mu, k(t)-1} - \sigma^{k(t)} \right\|^2 - \frac{\mu}{2} \left\| \pi^{\mu, k(t)} - \pi^{\mu, k(t)-1} \right\|^2$$

$$+ \frac{\mu}{2} \left\| \hat{\sigma}^{k(t)-1} - \hat{\sigma}^{k(t)} \right\|^2 + \frac{\mu}{2} \left\| \pi^{\mu, k(t)} - \pi^{\mu, k(t)-1} \right\|^2$$

$$- \frac{\mu}{2} \left\| \pi^{\mu, k(t)} - \pi^{\mu, k(t)-1} + \hat{\sigma}^{k(t)-1} + \hat{\sigma}^{k(t)} \right\|^2$$

$$= -\frac{\mu}{2} \left\| \pi^{\mu, k(t)} - \pi^{\mu, k(t)-1} \right\|^2 + \frac{\mu}{2} \left\| \hat{\sigma}^{k(t)} - \hat{\sigma}^{k(t)-1} \right\|^2$$

$$-\frac{\mu}{2}\left\|\pi^{\mu,k(t)}-\pi^{\mu,k(t)-1}+\hat{\sigma}^{k(t)-1}+\hat{\sigma}^{k(t)}\right\|^2$$

$$\leq-\frac{\mu}{2}\left\|\pi^{\mu,k(t)}-\pi^{\mu,k(t)-1}\right\|^2+\frac{\mu}{2}\left\|\hat{\sigma}^{k(t)}-\hat{\sigma}^{k(t)-1}\right\|^2$$

$$=-\frac{\mu}{2}\left\|\pi^{\mu,k(t)}-\pi^{\mu,k(t)-1}\right\|^2+\frac{\mu}{2}\left\|\hat{\sigma}^{k(t)}-\pi^{\mu,k(t)-1}+\pi^{\mu,k(t)-1}-\hat{\sigma}^{k(t)-1}\right\|^2$$

$$=\frac{\mu}{2}\left\|\pi^{\mu,k(t)-1}-\hat{\sigma}^{k(t)-1}\right\|^2+\frac{\mu}{2}\left\|\hat{\sigma}^{k(t)}-\pi^{\mu,k(t)-1}\right\|^2-\frac{\mu}{2}\left\|\pi^{\mu,k(t)}-\pi^{\mu,k(t)-1}\right\|^2$$

$$+\mu\left\langle\hat{\sigma}^{k(t)}-\pi^{\mu,k(t)-1},\pi^{\mu,k(t)-1}-\hat{\sigma}^{k(t)-1}\right\rangle. \tag{23}$$

Here, from the definition of $\hat{\sigma}^{k(t)}$, we have:

$$\frac{1}{2}\left\|\hat{\sigma}^{k(t)}-\pi^{\mu,k(t)-1}\right\|^2-\frac{1}{2}\left\|\pi^{\mu,k(t)}-\pi^{\mu,k(t)-1}\right\|^2$$

$$=\frac{1}{2}\left\|\frac{k(t)\sigma^{k(t)}+\sigma^1}{k(t)+1}-\pi^{\mu,k(t)-1}\right\|^2-\frac{1}{2}\left\|\pi^{\mu,k(t)}-\pi^{\mu,k(t)-1}\right\|^2$$

$$=\frac{1}{2}\left\langle\frac{k(t)\sigma^{k(t)}+\sigma^1}{k(t)+1}-\pi^{\mu,k(t)-1}+\pi^{\mu,k(t)}-\pi^{\mu,k(t)-1},\frac{k(t)\sigma^{k(t)}+\sigma^1}{k(t)+1}-\pi^{\mu,k(t)-1}-\pi^{\mu,k(t)}+\pi^{\mu,k(t)-1}\right\rangle$$

$$=\frac{1}{2}\left\langle\frac{\sigma^1+(k(t)+1)\pi^{\mu,k(t)}-2(k(t)+1)\pi^{\mu,k(t)-1}+k(t)\sigma^{k(t)}}{k(t)+1},\hat{\sigma}^{k(t)}-\pi^{\mu,k(t)}\right\rangle$$

$$=\frac{1}{2k(t)}\left\langle2(k(t)+1)\sigma^{k(t)+1}+2\sigma^1-2(k(t)+2)\pi^{\mu,k(t)},\hat{\sigma}^{k(t)}-\pi^{\mu,k(t)}\right\rangle$$

$$+\frac{1}{2k(t)}\left\langle-\frac{k(t)+2}{k(t)+1}\sigma^1+(3k(t)+4)\pi^{\mu,k(t)}-2(k(t)+1)\sigma^{k(t)+1}\hat{\sigma}^{k(t)}-\pi^{\mu,k(t)}\right\rangle$$

$$+\frac{1}{2k(t)}\left\langle-2k(t)\pi^{\mu,k(t)-1}+\frac{k(t)^2}{k(t)+1}\sigma^{k(t)},\hat{\sigma}^{k(t)}-\pi^{\mu,k(t)}\right\rangle$$

$$=\frac{k(t)+2}{k(t)}\left\langle\hat{\sigma}^{k(t)+1}-\pi^{\mu,k(t)},\hat{\sigma}^{k(t)}-\pi^{\mu,k(t)}\right\rangle$$

$$+\frac{1}{2k(t)}\left\langle-\frac{k(t)+2}{k(t)+1}\sigma^1-\frac{k(t)(k(t)+2)}{k(t)+1}\sigma^{k(t)}+(k(t)+2)\pi^{\mu,k(t)},\hat{\sigma}^{k(t)}-\pi^{\mu,k(t)}\right\rangle$$

$$+\frac{1}{2k(t)}\left\langle2(k(t)+1)(\pi^{\mu,k(t)}-\sigma^{k(t)+1})+2k(t)(\sigma^{k(t)}-\pi^{\mu,k(t)-1}),\hat{\sigma}^{k(t)}-\pi^{\mu,k(t)}\right\rangle$$

$$=-\frac{k(t)+2}{k(t)}\left\langle\hat{\sigma}^{k(t)+1}-\pi^{\mu,k(t)},\pi^{\mu,k(t)}-\hat{\sigma}^{k(t)}\right\rangle$$

$$-\frac{k(t)+2}{2k(t)}\left\langle\frac{k(t)\sigma^{k(t)}+\sigma^1}{k(t)+1}-\pi^{\mu,k(t)},\hat{\sigma}^{k(t)}-\pi^{\mu,k(t)}\right\rangle$$

$$+\frac{1}{k(t)}\left\langle(k(t)+1)(\pi^{\mu,k(t)}-\sigma^{k(t)+1})+k(t)(\sigma^{k(t)}-\pi^{\mu,k(t)-1}),\hat{\sigma}^{k(t)}-\pi^{\mu,k(t)}\right\rangle$$

$$=-\frac{k(t)+2}{k(t)}\left\langle\hat{\sigma}^{k(t)+1}-\pi^{\mu,k(t)},\pi^{\mu,k(t)}-\hat{\sigma}^{k(t)}\right\rangle-\frac{k(t)+2}{2k(t)}\left\|\hat{\sigma}^{k(t)}-\pi^{\mu,k(t)}\right\|^2$$

$$+\frac{1}{k(t)}\left\langle(k(t)+1)(\pi^{\mu,k(t)}-\sigma^{k(t)+1})+k(t)(\sigma^{k(t)}-\pi^{\mu,k(t)-1}),\hat{\sigma}^{k(t)}-\pi^{\mu,k(t)}\right\rangle. \tag{24}$$

Combining Eq. (23) and Eq. (24) yields for any $t\geq1$ such that $k(t)\geq2$:

$$\frac{k(t)+2}{2k(t)}\left\|\pi^{\mu,k(t)}-\hat{\sigma}^{k(t)}\right\|^2+\frac{k(t)+2}{k(t)}\left\langle\hat{\sigma}^{k(t)+1}-\pi^{\mu,k(t)},\pi^{\mu,k(t)}-\hat{\sigma}^{k(t)}\right\rangle$$

$$\leq\frac{1}{2}\left\|\pi^{\mu,k(t)-1}-\hat{\sigma}^{k(t)-1}\right\|^2+\left\langle\hat{\sigma}^{k(t)}-\pi^{\mu,k(t)-1},\pi^{\mu,k(t)-1}-\hat{\sigma}^{k(t)-1}\right\rangle$$

$$+\frac{1}{k(t)}\left\langle(k(t)+1)(\pi^{\mu,k(t)}-\sigma^{k(t)+1})+k(t)(\sigma^{k(t)}-\pi^{\mu,k(t)-1}),\hat{\sigma}^{k(t)}-\pi^{\mu,k(t)}\right\rangle.$$

Multiplying both sides by $k(t)(k(t)+1)$, we have:

$$\frac{(k(t)+1)(k(t)+2)}{2}\left\|\pi^{\mu,k(t)}-\hat{\sigma}^{k(t)}\right\|^2 + (k(t)+1)(k(t)+2)\left\langle\hat{\sigma}^{k(t)+1}-\pi^{\mu,k(t)},\pi^{\mu,k(t)}-\hat{\sigma}^{k(t)}\right\rangle$$

$$\leq \frac{k(t)(k(t)+1)}{2}\left\|\pi^{\mu,k(t)-1}-\hat{\sigma}^{k(t)-1}\right\|^2 + k(t)(k(t)+1)\left\langle\hat{\sigma}^{k(t)}-\pi^{\mu,k(t)-1},\pi^{\mu,k(t)-1}-\hat{\sigma}^{k(t)-1}\right\rangle$$

$$+ (k(t)+1)\left\langle(k(t)+1)(\pi^{\mu,k(t)}-\sigma^{k(t)+1})+k(t)(\sigma^{k(t)}-\pi^{\mu,k(t)-1}),\hat{\sigma}^{k(t)}-\pi^{\mu,k(t)}\right\rangle.$$

$\square$

## C  PROOFS FOR THEOREM 4.3

### C.1  PROOF OF THEOREM 4.3

*Proof of Theorem 4.3.* Let us define $K := \frac{T}{T_\sigma}$. We can decompose the gap function for $\pi^{T+1}$ as follows:

$$\text{GAP}(\pi^{T+1})$$
$$= \max_{x\in\mathcal{X}}\left\langle V(\pi^{T+1}),x-\pi^{T+1}\right\rangle$$
$$= \max_{x\in\mathcal{X}}\left(\left\langle V(\pi^{\mu,K}),x-\pi^{\mu,K}\right\rangle - \left\langle V(\pi^{\mu,K}),x-\pi^{\mu,K}\right\rangle + \left\langle V(\pi^{T+1}),x-\pi^{T+1}\right\rangle\right)$$
$$= \max_{x\in\mathcal{X}}\left(\left\langle V(\pi^{\mu,K}),x-\pi^{\mu,K}\right\rangle - \left\langle V(\pi^{\mu,K})-V(\pi^{T+1}),x-\pi^{T+1}\right\rangle + \left\langle V(\pi^{\mu,K}),\pi^{\mu,K}-\pi^{T+1}\right\rangle\right)$$
$$\leq \max_{x\in\mathcal{X}}\left(\left\langle V(\pi^{\mu,K}),x-\pi^{\mu,K}\right\rangle + D\left\|V(\pi^{\mu,K})-V(\pi^{T+1})\right\| + \zeta\left\|\pi^{\mu,K}-\pi^{T+1}\right\|\right)$$
$$\leq \text{GAP}(\pi^{\mu,K}) + (LD+\zeta)\left\|\pi^{\mu,K}-\pi^{T+1}\right\|$$
$$\leq D\cdot\min_{c\in N_{\mathcal{X}}(\pi^{\mu,K})}\left\|-V(\pi^{\mu,K})+c\right\| + (LD+\zeta)\left\|\pi^{\mu,K}-\pi^{T+1}\right\|,$$

where the last inequality follows from Lemma 2.3. From the first-order optimality condition for $\pi^{\mu,K}$, we have for any $x\in\mathcal{X}$:

$$\left\langle V(\pi^{\mu,K})-\mu\left(\pi^{\mu,K}-\frac{K\sigma^K+\sigma^1}{K+1}\right),\pi^{\mu,K}-x\right\rangle \geq 0,$$

and then $V(\pi^{\mu,K})-\mu\left(\pi^{\mu,K}-\frac{K\sigma^K+\sigma^1}{K+1}\right)\in N_{\mathcal{X}}(\pi^{\mu,K})$. Thus, the gap function for $\pi^{T+1}$ can be bounded by:

$$\text{GAP}(\pi^{T+1}) \leq \mu D\cdot\left\|\pi^{\mu,K}-\frac{K\sigma^K+\sigma^1}{K+1}\right\| + (LD+\zeta)\left\|\pi^{\mu,K}-\pi^{T+1}\right\|$$
$$= \mu D\cdot\left\|\frac{\sigma^K-\sigma^1}{K+1}+\pi^{\mu,K}-\sigma^K\right\| + (LD+\zeta)\left\|\pi^{\mu,K}-\pi^{T+1}\right\|$$
$$\leq \mu D\cdot\left(\frac{D}{K+1}+\left\|\pi^{\mu,K}-\sigma^K\right\|\right) + (LD+\zeta)\left\|\pi^{\mu,K}-\pi^{T+1}\right\|.$$

Taking its expectation yields:

$$\mathbb{E}\left[\text{GAP}(\pi^{T+1})\right] \leq \frac{\mu D^2}{K+1} + \mu D\cdot\mathbb{E}\left[\left\|\pi^{\mu,K}-\sigma^K\right\|\right] + (LD+\zeta)\cdot\mathbb{E}\left[\left\|\pi^{\mu,K}-\pi^{T+1}\right\|\right]$$
$$\leq \frac{\mu D^2}{K+1} + \mu D\cdot\mathbb{E}\left[\left\|\pi^{\mu,K}-\sigma^K\right\|\right] + (LD+\zeta)\cdot\sqrt{\mathbb{E}\left[\left\|\pi^{\mu,K}-\pi^{T+1}\right\|^2\right]}.$$
$$(25)$$

Here, we derive the following upper bound on $\mathbb{E}\left[\left\|\pi^{\mu,k(t)}-\pi^{t+1}\right\|^2\right]$:

**Lemma C.1.** *Let $\kappa = \frac{\mu}{2}, \theta = \frac{3\mu^2 + 8L^2}{2\mu}$. Suppose that Assumption 4.2 holds. If we set $\eta_t = \frac{1}{\kappa(t - T_\sigma(k(t)-1)) + 2\theta}$, we have for any $t \geq 1$:*

$$\mathbb{E}\left[\left\|\pi^{\mu,k(t)} - \pi^{t+1}\right\|^2\right] \leq \frac{2\theta}{\kappa\left(t - (k(t)-1)T_\sigma\right) + 2\theta}\left(D^2 + \frac{C^2}{\kappa\theta}\ln\left(\frac{\kappa\left(t - (k(t)-1)T_\sigma\right)}{2\theta} + 1\right)\right).$$

Setting $t = T = KT_\sigma$, we can write $k(t) = \lfloor\frac{KT_\sigma - 1}{T_\sigma}\rfloor + 1 = K$. Therefore, from Lemma C.1, we have:

$$\mathbb{E}\left[\left\|\pi^{\mu,K} - \pi^{T+1}\right\|^2\right] \leq \frac{2\theta}{\kappa T_\sigma + 2\theta}\left(D^2 + \frac{C^2}{\kappa\theta}\ln\left(\frac{\kappa T_\sigma}{2\theta} + 1\right)\right). \tag{26}$$

On the other hand, in terms of $\mathbb{E}\left[\left\|\pi^{\mu,k(t)} - \sigma^{k(t)}\right\|\right]$, we introduce the following lemma:

**Lemma C.2.** *If we set $\eta_t = \frac{1}{\kappa(t - T_\sigma(k(t)-1)) + 2\theta}$ and $T_\sigma \geq \max(1, T^{\frac{6}{7}})$, we have for any $t \geq 1$:*

$$\mathbb{E}\left[\left\|\pi^{\mu,k(t)} - \sigma^{k(t)}\right\|\right] \leq \frac{6\left(\sqrt{\kappa} + \sqrt{\theta} + \sqrt{D\theta} + \sqrt{D}\right)}{k(t)}\left(\sqrt{\frac{1}{\kappa}\left(D^2 + \frac{C^2}{\kappa\theta}\ln\left(\frac{\kappa T}{2\theta} + 1\right)\right)} + 1\right).$$

By setting $t = KT_\sigma$ in this lemma, we get:

$$\mathbb{E}\left[\left\|\pi^{\mu,K} - \sigma^{K}\right\|\right] \leq \frac{6\left(\sqrt{\kappa} + \sqrt{\theta} + \sqrt{D\theta} + \sqrt{D}\right)}{K}\left(\sqrt{\frac{1}{\kappa}\left(D^2 + \frac{C^2}{\kappa\theta}\ln\left(\frac{\kappa T}{2\theta} + 1\right)\right)} + 1\right). \tag{27}$$

Combining Eq. (25), Eq. (26), and Eq. (27), we have:

$$\mathbb{E}\left[\text{GAP}(\sigma^{K+1})\right]$$

$$\leq \frac{\mu D^2}{K+1} + \mu D \cdot \frac{6\left(\sqrt{\kappa} + \sqrt{\theta} + \sqrt{D\theta} + \sqrt{D}\right)}{K}\left(\sqrt{\frac{1}{\kappa}\left(D^2 + \frac{C^2}{\kappa\theta}\ln\left(\frac{\kappa T}{2\theta} + 1\right)\right)} + 1\right)$$

$$+ (LD + \zeta)\cdot\sqrt{\frac{2\theta}{\kappa T_\sigma + 2\theta}\left(D^2 + \frac{C^2}{\kappa\theta}\ln\left(\frac{\kappa T_\sigma}{2\theta} + 1\right)\right)}$$

$$\leq \mu D^2 \frac{T_\sigma}{T} + \mu D \cdot \frac{6T_\sigma\left(\sqrt{\kappa} + \sqrt{\theta} + \sqrt{D\theta} + \sqrt{D}\right)}{T}\left(\sqrt{\frac{1}{\kappa}\left(D^2 + \frac{C^2}{\kappa\theta}\ln\left(\frac{\kappa T}{2\theta} + 1\right)\right)} + 1\right)$$

$$+ (LD + \zeta)\cdot\sqrt{\frac{2\theta}{\kappa T_\sigma}\left(D^2 + \frac{C^2}{\kappa\theta}\ln\left(\frac{\kappa T}{2\theta} + 1\right)\right)},$$

where the second inequality follows from $K = \frac{T}{T_\sigma}$. Finally, since $T_\sigma = c \cdot \max(1, T^{\frac{6}{7}})$, we have for any $T \geq T_\sigma$:

$$\mathbb{E}\left[\text{GAP}(\sigma^{K+1})\right]$$

$$\leq \frac{c\mu D^2}{T^{\frac{1}{7}}} + \frac{6c\mu D\left(\sqrt{\kappa} + \sqrt{\theta} + \sqrt{D\theta} + \sqrt{D}\right)}{T^{\frac{1}{7}}}\left(\sqrt{\frac{1}{\kappa}\left(D^2 + \frac{C^2}{\kappa\theta}\ln\left(\frac{\kappa T}{2\theta} + 1\right)\right)} + 1\right)$$

$$+ \frac{(LD + \zeta)}{T^{\frac{3}{7}}}\sqrt{\frac{2\theta}{\kappa}\left(D^2 + \frac{C^2}{\kappa\theta}\ln\left(\frac{\kappa T}{2\theta} + 1\right)\right)}$$

$$\leq \frac{6c\mu D\left(\sqrt{\kappa} + \sqrt{\theta} + \sqrt{D\theta} + \sqrt{D} + D\right)}{T^{\frac{1}{7}}}\left(\sqrt{\frac{1}{\kappa}\left(D^2 + \frac{C^2}{\kappa\theta}\ln\left(\frac{\kappa T}{2\theta} + 1\right)\right)} + 1\right)$$

$$+ \frac{(LD + \zeta)\sqrt{2\theta}}{T^{\frac{1}{7}}} \left( \sqrt{\frac{1}{\kappa} \left( D^2 + \frac{C^2}{\kappa\theta} \ln\left(\frac{\kappa T}{2\theta} + 1\right) \right)} + 1 \right)$$

$$\leq \frac{9c \left( \mu D + LD + \zeta \right) \left( \sqrt{\kappa} + \sqrt{\theta} + \sqrt{D\theta} + \sqrt{D} + D \right)}{T^{\frac{1}{7}}} \left( \sqrt{\frac{1}{\kappa} \left( D^2 + \frac{C^2}{\kappa\theta} \ln\left(\frac{\kappa T}{2\theta} + 1\right) \right)} + 1 \right).$$

Since $T = T_\sigma K$, we have finally:

$$\mathbb{E}\left[ \mathrm{GAP}(\pi^{T+1}) \right]$$

$$\leq \frac{9c \left( \mu D + LD + \zeta \right) \left( \sqrt{\kappa} + \sqrt{\theta} + \sqrt{D\theta} + \sqrt{D} + D \right)}{T^{\frac{1}{7}}} \left( \sqrt{\frac{1}{\kappa} \left( D^2 + \frac{C^2}{\kappa\theta} \ln\left(\frac{\kappa T}{2\theta} + 1\right) \right)} + 1 \right)$$

$$= \frac{9c \left( D(\mu + L) + \zeta \right) \left( \sqrt{\kappa} + (\sqrt{D} + 1)(\sqrt{D} + \sqrt{\theta}) \right)}{T^{\frac{1}{7}}} \left( \sqrt{\frac{1}{\kappa} \left( D^2 + \frac{C^2}{\kappa\theta} \ln\left(\frac{\kappa T}{2\theta} + 1\right) \right)} + 1 \right)$$

$$\leq \frac{18c \left( D(\mu + L) + \zeta \right) \left( \sqrt{\kappa} + \sqrt{(D+1)(D+\theta)} \right)}{T^{\frac{1}{7}}} \left( \sqrt{\frac{1}{\kappa} \left( D^2 + \frac{C^2}{\kappa\theta} \ln\left(\frac{\kappa T}{2\theta} + 1\right) \right)} + 1 \right)$$

$$\leq \frac{26c \left( D(\mu + L) + \zeta \right) \sqrt{(D+1)(D+\theta) + \kappa}}{T^{\frac{1}{7}}} \left( \sqrt{\frac{1}{\kappa} \left( D^2 + \frac{C^2}{\kappa\theta} \ln\left(\frac{\kappa T}{2\theta} + 1\right) \right)} + 1 \right).$$

$\square$

## C.2 PROOF OF LEMMA C.1

*Proof of Lemma C.1.* From the first-order optimality condition for $\pi^{t+1}$, we have for $t \geq 1$:

$$\left\langle \eta_t \left( \hat{V}(\pi^t) - \mu(\pi^t - \hat{\sigma}^{k(t)}) \right) - \pi^{t+1} + \pi^t, \pi^{t+1} - \pi^{\mu, k(t)} \right\rangle \geq 0. \tag{28}$$

Combining Eq. (28), Eq. (12), and Eq. (14), we have:

$$\frac{1}{2} \left\| \pi^{\mu, k(t)} - \pi^{t+1} \right\|^2 - \frac{1}{2} \left\| \pi^{\mu, k(t)} - \pi^t \right\|^2 + \frac{1}{2} \left\| \pi^{t+1} - \pi^t \right\|^2$$

$$\leq \eta_t \left\langle \hat{V}(\pi^t) - \mu(\pi^t - \hat{\sigma}^{k(t)}), \pi^{t+1} - \pi^{\mu, k(t)} \right\rangle$$

$$= \eta_t \left\langle V(\pi^{t+1}) - \mu(\pi^{t+1} - \hat{\sigma}^{k(t)}), \pi^{t+1} - \pi^{\mu, k(t)} \right\rangle + \eta_t \left\langle \hat{V}(\pi^t) - V(\pi^{t+1}) - \mu(\pi^t - \pi^{t+1}), \pi^{t+1} - \pi^{\mu, k(t)} \right\rangle$$

$$\leq \eta_t \left\langle V(\pi^{\mu, k(t)}) - \mu(\pi^{t+1} - \hat{\sigma}^{k(t)}), \pi^{t+1} - \pi^{\mu, k(t)} \right\rangle + \eta_t \left\langle \hat{V}(\pi^t) - V(\pi^{t+1}) - \mu(\pi^t - \pi^{t+1}), \pi^{t+1} - \pi^{\mu, k(t)} \right\rangle$$

$$= \eta_t \left\langle V(\pi^{\mu, k(t)}) - \mu(\pi^{\mu, k(t)} - \hat{\sigma}^{k(t)}), \pi^{t+1} - \pi^{\mu, k(t)} \right\rangle - \eta_t \mu \left\| \pi^{\mu, k(t)} - \pi^{t+1} \right\|^2$$

$$\quad + \eta_t \left\langle V(\pi^t) - V(\pi^{t+1}), \pi^{t+1} - \pi^{\mu, k(t)} \right\rangle - \eta_t \mu \left\langle \pi^t - \pi^{t+1}, \pi^{t+1} - \pi^{\mu, k(t)} \right\rangle + \eta_t \left\langle \xi^t, \pi^{t+1} - \pi^{\mu, k(t)} \right\rangle$$

$$\leq -\eta_t \mu \left\| \pi^{\mu, k(t)} - \pi^{t+1} \right\|^2 + \eta_t \mu \left\langle \pi^{t+1} - \pi^t, \pi^{t+1} - \pi^{\mu, k(t)} \right\rangle$$

$$\quad + \eta_t \left\langle V(\pi^t) - V(\pi^{t+1}), \pi^{t+1} - \pi^{\mu, k(t)} \right\rangle + \eta_t \left\langle \xi^t, \pi^{t+1} - \pi^{\mu, k(t)} \right\rangle$$

$$= -\eta_t \mu \left\| \pi^{\mu, k(t)} - \pi^{t+1} \right\|^2 + \frac{\eta_t \mu}{2} \left\| \pi^{t+1} - \pi^t \right\|^2 + \frac{\eta_t \mu}{2} \left\| \pi^{t+1} - \pi^{\mu, k(t)} \right\|^2 - \frac{\eta_t \mu}{2} \left\| \pi^t - \pi^{\mu, k(t)} \right\|^2$$

$$\quad + \eta_t \left\langle V(\pi^t) - V(\pi^{t+1}), \pi^{t+1} - \pi^{\mu, k(t)} \right\rangle + \eta_t \left\langle \xi^t, \pi^{t+1} - \pi^{\mu, k(t)} \right\rangle$$

$$= -\frac{\eta_t \mu}{2} \left\| \pi^{t+1} - \pi^{\mu, k(t)} \right\|^2 - \frac{\eta_t \mu}{2} \left\| \pi^t - \pi^{\mu, k(t)} \right\|^2 + \frac{\eta_t \mu}{2} \left\| \pi^{t+1} - \pi^t \right\|^2$$

$$\quad + \eta_t \left\langle V(\pi^t) - V(\pi^{t+1}), \pi^{t+1} - \pi^{\mu, k(t)} \right\rangle + \eta_t \left\langle \xi^t, \pi^{t+1} - \pi^{\mu, k(t)} \right\rangle, \tag{29}$$

where the third inequality follows from Eq. (1). From Cauchy-Schwarz inequality and Young's inequality, the fourth term on the right-hand side of this inequality can be bounded by:

$$
\left\langle V(\pi^t) - V(\pi^{t+1}), \pi^{t+1} - \pi^{\mu,k(t)} \right\rangle
$$

$$
\leq \left\| V(\pi^t) - V(\pi^{t+1}) \right\| \cdot \left\| \pi^{t+1} - \pi^{\mu,k(t)} \right\|
$$

$$
\leq L \left\| \pi^t - \pi^{t+1} \right\| \cdot \left\| \pi^{t+1} - \pi^{\mu,k(t)} \right\|
$$

$$
\leq \frac{2L^2}{\mu} \left\| \pi^t - \pi^{t+1} \right\|^2 + \frac{\mu}{8} \left\| \pi^{t+1} - \pi^{\mu,k(t)} \right\|^2
$$

$$
\leq \frac{2L^2}{\mu} \left\| \pi^t - \pi^{t+1} \right\|^2 + \frac{\mu}{4} \left\| \pi^t - \pi^{\mu,k(t)} \right\|^2 + \frac{\mu}{4} \left\| \pi^{t+1} - \pi^t \right\|^2
$$

$$
= \left( \frac{4L^2}{\mu} + \frac{\mu}{2} \right) \frac{\left\| \pi^t - \pi^{t+1} \right\|^2}{2} + \frac{\mu}{2} \frac{\left\| \pi^t - \pi^{\mu,k(t)} \right\|^2}{2}. \tag{30}
$$

By combining Eq. (29) and Eq. (30), we have:

$$
\left\| \pi^{\mu,k(t)} - \pi^{t+1} \right\|^2 \leq -\eta_t \mu \left\| \pi^{t+1} - \pi^{\mu,k(t)} \right\|^2 + \left( 1 - \frac{\eta_t \mu}{2} \right) \left\| \pi^t - \pi^{\mu,k(t)} \right\|^2
$$

$$
- \left( 1 - \eta_t \left( \frac{3\mu}{2} + \frac{4L^2}{\mu} \right) \right) \left\| \pi^{t+1} - \pi^t \right\|^2 + 2\eta_t \left\langle \xi^t, \pi^{t+1} - \pi^{\mu,k(t)} \right\rangle
$$

$$
\leq \left( 1 - \frac{\eta_t \mu}{2} \right) \left\| \pi^t - \pi^{\mu,k(t)} \right\|^2 - \left( 1 - \eta_t \left( \frac{3\mu}{2} + \frac{4L^2}{\mu} \right) \right) \left\| \pi^{t+1} - \pi^t \right\|^2
$$

$$
+ 2\eta_t \left\langle \xi^t, \pi^t - \pi^{\mu,k(t)} \right\rangle + 2\eta_t \left\langle \xi^t, \pi^{t+1} - \pi^t \right\rangle
$$

$$
= (1 - \eta_t \kappa) \left\| \pi^t - \pi^{\mu,k(t)} \right\|^2 - (1 - \eta_t \theta) \left\| \pi^{t+1} - \pi^t \right\|^2
$$

$$
+ 2\eta_t \left\langle \xi^t, \pi^t - \pi^{\mu,k(t)} \right\rangle + 2\eta_t \left\langle \xi^t, \pi^{t+1} - \pi^t \right\rangle.
$$

By taking the expectation conditioned on $\mathcal{F}_t$ for both sides and using Assumption 4.2 (a) and (b),

$$
\mathbb{E} \left[ \left\| \pi^{\mu,k(t)} - \pi^{t+1} \right\|^2 \mid \mathcal{F}_t \right]
$$

$$
\leq (1 - \eta_t \kappa) \mathbb{E} \left[ \left\| \pi^t - \pi^{\mu,k(t)} \right\|^2 \mid \mathcal{F}_t \right] - (1 - \eta_t \theta) \mathbb{E} \left[ \left\| \pi^{t+1} - \pi^t \right\|^2 \mid \mathcal{F}_t \right]
$$

$$
+ 2\eta_t \left\langle \mathbb{E} \left[ \xi^t \mid \mathcal{F}_t \right], \pi^t - \pi^{\mu,k(t)} \right\rangle + 2\eta_t \mathbb{E} \left[ \left\langle \xi^t, \pi^{t+1} - \pi^t \right\rangle \mid \mathcal{F}_t \right]
$$

$$
= (1 - \eta_t \kappa) \left\| \pi^t - \pi^{\mu,k(t)} \right\|^2 - (1 - \eta_t \theta) \mathbb{E} \left[ \left\| \pi^{t+1} - \pi^t \right\|^2 \mid \mathcal{F}_t \right] + 2\eta_t \mathbb{E} \left[ \left\langle \xi^t, \pi^{t+1} - \pi^t \right\rangle \mid \mathcal{F}_t \right]
$$

$$
\leq (1 - \eta_t \kappa) \left\| \pi^t - \pi^{\mu,k(t)} \right\|^2 - (1 - \eta_t \theta) \mathbb{E} \left[ \left\| \pi^{t+1} - \pi^t \right\|^2 \mid \mathcal{F}_t \right]
$$

$$
+ \frac{\eta_t^2}{1 - \eta_t \theta} \mathbb{E} \left[ \left\| \xi^t \right\|^2 \mid \mathcal{F}_t \right] + (1 - \eta_t \theta) \mathbb{E} \left[ \left\| \pi^{t+1} - \pi^t \right\|^2 \mid \mathcal{F}_t \right]
$$

$$
\leq (1 - \eta_t \kappa) \left\| \pi^t - \pi^{\mu,k(t)} \right\|^2 + \frac{\eta_t^2}{1 - \eta_t \theta} \mathbb{E} \left[ \left\| \xi^t \right\|^2 \mid \mathcal{F}_t \right]
$$

$$
\leq (1 - \eta_t \kappa) \left\| \pi^t - \pi^{\mu,k(t)} \right\|^2 + 2\eta_t^2 \mathbb{E} \left[ \left\| \xi^t \right\|^2 \mid \mathcal{F}_t \right]
$$

$$
\leq (1 - \eta_t \kappa) \left\| \pi^t - \pi^{\mu,k(t)} \right\|^2 + 2\eta_t^2 C^2.
$$

Therefore, under the setting where $\eta_t = \frac{1}{\kappa(t - T_\sigma(k(t)-1)) + 2\theta}$, we have for any $t \geq 1$:

$$
\mathbb{E} \left[ \left\| \pi^{\mu,k(t)} - \pi^{t+1} \right\|^2 \mid \mathcal{F}_t \right] \leq \left( 1 - \frac{1}{t - T_\sigma(k(t)-1) + 2\theta/\kappa} \right) \left\| \pi^t - \pi^{\mu,k(t)} \right\|^2 + 2\eta_t^2 C^2.
$$

Rearranging and taking the expectations, we get:

$$(t - T_\sigma(k(t) - 1) + 2\theta/\kappa) \, \mathbb{E} \left[ \left\| \pi^{\mu,k(t)} - \pi^{t+1} \right\|^2 \right]$$

$$\leq (t - 1 - T_\sigma(k(t) - 1) + 2\theta/\kappa) \, \mathbb{E} \left[ \left\| \pi^{\mu,k(t)} - \pi^t \right\|^2 \right] + \frac{2C^2}{\kappa \left( \kappa(t - T_\sigma(k(t) - 1)) + 2\theta \right)}.$$

Since $k(s) = k(t)$ for any $s \in [(k(t) - 1)T_\sigma + 1, T]$, telescoping the sum yields:

$$(t - T_\sigma(k(t) - 1) + 2\theta/\kappa) \, \mathbb{E} \left[ \left\| \pi^{\mu,k(t)} - \pi^{t+1} \right\|^2 \right]$$

$$\leq (s - 1 - T_\sigma(k(t) - 1) + 2\theta/\kappa) \, \mathbb{E} \left[ \left\| \pi^{\mu,k(t)} - \pi^s \right\|^2 \right] + \sum_{m=s}^{t} \frac{2C^2}{\kappa \left( \kappa(m - T_\sigma(k(t) - 1)) + 2\theta \right)}.$$

Defining $s = (k(t) - 1)T_\sigma + 1$,

$$(t - T_\sigma(k(t) - 1) + 2\theta/\kappa) \, \mathbb{E} \left[ \left\| \pi^{\mu,k(t)} - \pi^{t+1} \right\|^2 \right]$$

$$\leq \frac{2\theta}{\kappa} \mathbb{E} \left[ \left\| \pi^{\mu,k(t)} - \pi^{(k(t)-1)T_\sigma+1} \right\|^2 \right] + \frac{2C^2}{\kappa} \sum_{m=(k(t)-1)T_\sigma+1}^{t} \frac{1}{\kappa(m - T_\sigma(k(t) - 1)) + 2\theta}.$$

Therefore,

$$\mathbb{E} \left[ \left\| \pi^{\mu,k(t)} - \pi^{t+1} \right\|^2 \right] \leq \frac{2\theta}{\kappa \left( t - T_\sigma(k(t) - 1) \right) + 2\theta} \mathbb{E} \left[ \left\| \pi^{\mu,k(t)} - \pi^{(k(t)-1)T_\sigma+1} \right\|^2 \right]$$

$$+ \frac{2C^2}{\kappa \left( t - T_\sigma(k(t) - 1) \right) + 2\theta} \sum_{m=1}^{t-(k(t)-1)T_\sigma} \frac{1}{\kappa m + 2\theta}. \tag{31}$$

Here, we have:

$$\sum_{m=1}^{t-(k(t)-1)T_\sigma} \frac{1}{\kappa m + 2\theta} \leq \int_0^{t-(k(t)-1)T_\sigma} \frac{1}{\kappa x + 2\theta} dx = \frac{1}{\kappa} \ln \left( \frac{\kappa \left( t - (k(t) - 1)T_\sigma \right)}{2\theta} + 1 \right). \tag{32}$$

Combining Eq. (31), Eq. (32), and the fact that $\pi^{(k(t)-1)T_\sigma+1} = \sigma^{k(t)}$, we have:

$$\mathbb{E} \left[ \left\| \pi^{\mu,k(t)} - \pi^{t+1} \right\|^2 \right]$$

$$\leq \frac{2\theta}{\kappa \left( t - (k(t) - 1)T_\sigma \right) + 2\theta} \left( \mathbb{E} \left[ \left\| \pi^{\mu,k(t)} - \sigma^{k(t)} \right\|^2 \right] + \frac{C^2}{\kappa\theta} \ln \left( \frac{\kappa \left( t - (k(t) - 1)T_\sigma \right)}{2\theta} + 1 \right) \right)$$

$$\leq \frac{2\theta}{\kappa \left( t - (k(t) - 1)T_\sigma \right) + 2\theta} \left( D^2 + \frac{C^2}{\kappa\theta} \ln \left( \frac{\kappa \left( t - (k(t) - 1)T_\sigma \right)}{2\theta} + 1 \right) \right).$$

$\square$

## C.3   PROOF OF LEMMA C.2

*Proof of Lemma C.2.*  First, from Lemma C.1, we have for any $k \geq 1$:

$$\mathbb{E} \left[ \left\| \pi^{\mu,k} - \sigma^{k+1} \right\|^2 \right] \leq \frac{2\theta}{\kappa T_\sigma + 2\theta} \left( D^2 + \frac{C^2}{\kappa\theta} \ln \left( \frac{\kappa T_\sigma}{2\theta} + 1 \right) \right).$$

Moreover, by taking the expectation of Eq. (21), we have for any $t \geq 1$ such that $k(t) \geq 2$:

$$\mathbb{E} \left[ \left\| \pi^{\mu,k(t)} - \sigma^{k(t)} \right\|^2 \right] \leq \frac{2D}{k(t) + 1} \mathbb{E} \left[ \left\| \pi^{\mu,k(t)} - \sigma^{k(t)} \right\| \right] + \frac{12D^2}{(k(t) + 1)^2}$$

$$+ 8\mathbb{E} \left[ \left\| \sigma^{k(t)+1} - \pi^{\mu,k(t)} \right\|^2 \right] + 8D \sum_{l=1}^{k(t)} \mathbb{E} \left[ \left\| \pi^{\mu,l} - \sigma^{l+1} \right\| \right].$$

Combining these inequalities, we get for any $t \geq 1$ such that $k(t) \geq 2$:

$$\mathbb{E}\left[\left\|\pi^{\mu,k(t)} - \sigma^{k(t)}\right\|^2\right] \leq \frac{2D}{k(t)+1}\mathbb{E}\left[\left\|\pi^{\mu,k(t)} - \sigma^{k(t)}\right\|\right] + \frac{12D^2}{(k(t)+1)^2}$$
$$+ \frac{16\theta}{\kappa T_\sigma}\left(D^2 + \frac{C^2}{\kappa\theta}\ln\left(\frac{\kappa T_\sigma}{2\theta}+1\right)\right) + 8Dk(t)\sqrt{\frac{2\theta}{\kappa T_\sigma}\left(D^2 + \frac{C^2}{\kappa\theta}\ln\left(\frac{\kappa T_\sigma}{2\theta}+1\right)\right)}.$$

Since $T_\sigma \geq \max(1, T^{\frac{6}{7}}) \Rightarrow \frac{k(t)^3}{\sqrt{T_\sigma}} \leq 1$, we have:

$$\mathbb{E}\left[\left(\left\|\pi^{\mu,k(t)} - \sigma^{k(t)}\right\| - \frac{D}{k(t)+1}\right)^2\right] \leq \frac{13D^2}{k(t)^2} + \frac{16\theta}{\kappa k(t)^2}\left(D^2 + \frac{C^2}{\kappa\theta}\ln\left(\frac{\kappa T}{2\theta}+1\right)\right)$$
$$+ \frac{8D}{k(t)^2}\sqrt{\frac{2\theta}{\kappa}\left(D^2 + \frac{C^2}{\kappa\theta}\ln\left(\frac{\kappa T}{2\theta}+1\right)\right)}.$$

Since $\mathbb{E}[X]^2 \leq \mathbb{E}[X^2]$ for any random variable $X$, we get:

$$\frac{13D^2}{k(t)^2} + \frac{16\theta}{\kappa k(t)^2}\left(D^2 + \frac{C^2}{\kappa\theta}\ln\left(\frac{\kappa T}{2\theta}+1\right)\right) + \frac{8D}{k(t)^2}\sqrt{\frac{2\theta}{\kappa}\left(D^2 + \frac{C^2}{\kappa\theta}\ln\left(\frac{\kappa T}{2\theta}+1\right)\right)}$$
$$\geq \mathbb{E}\left[\left(\left\|\pi^{\mu,k(t)} - \sigma^{k(t)}\right\| - \frac{D}{k(t)+1}\right)^2\right]$$
$$\geq \mathbb{E}\left[\left\|\pi^{\mu,k(t)} - \sigma^{k(t)}\right\| - \frac{D}{k(t)+1}\right]^2$$
$$= \left(\mathbb{E}\left[\left\|\pi^{\mu,k(t)} - \sigma^{k(t)}\right\|\right] - \frac{D}{k(t)+1}\right)^2.$$

Then, we have:

$$\mathbb{E}\left[\left\|\pi^{\mu,k(t)} - \sigma^{k(t)}\right\|\right]$$
$$\leq \frac{D}{k(t)} + \frac{4D}{k(t)} + \frac{4\sqrt{\theta}}{\sqrt{\kappa}k(t)}\sqrt{D^2 + \frac{C^2}{\kappa\theta}\ln\left(\frac{\kappa T}{2\theta}+1\right)} + \frac{3\sqrt{D}}{k(t)}\left(\frac{2\theta}{\kappa}\left(D^2 + \frac{C^2}{\kappa\theta}\ln\left(\frac{\kappa T}{2\theta}+1\right)\right)\right)^{\frac{1}{4}}$$
$$\leq \frac{5(\sqrt{\kappa}+\sqrt{\theta})}{k(t)\sqrt{\kappa}}\sqrt{D^2 + \frac{C^2}{\kappa\theta}\ln\left(\frac{\kappa T}{2\theta}+1\right)} + \frac{6\sqrt{D}(\sqrt{\theta}+1)}{k(t)}\left(\sqrt{\frac{1}{\kappa}\left(D^2 + \frac{C^2}{\kappa\theta}\ln\left(\frac{\kappa T}{2\theta}+1\right)\right)} + 1\right)$$
$$\leq \frac{6\left(\sqrt{\kappa}+\sqrt{\theta}+\sqrt{D\theta}+\sqrt{D}\right)}{k(t)}\left(\sqrt{\frac{1}{\kappa}\left(D^2 + \frac{C^2}{\kappa\theta}\ln\left(\frac{\kappa T}{2\theta}+1\right)\right)} + 1\right).$$

Furthermore, for $k(t) = 1$, we have:

$$\mathbb{E}\left[\left\|\pi^{\mu,1} - \sigma^1\right\|\right] \leq D \leq \frac{6\left(\sqrt{\kappa}+\sqrt{\theta}+\sqrt{D\theta}+\sqrt{D}\right)}{1}\left(\sqrt{\frac{1}{\kappa}\left(D^2 + \frac{C^2}{\kappa\theta}\ln\left(\frac{\kappa T}{2\theta}+1\right)\right)} + 1\right).$$

Therefore, we have for any $t \geq 1$:

$$\mathbb{E}\left[\left\|\pi^{\mu,k(t)} - \sigma^{k(t)}\right\|\right] \leq \frac{6\left(\sqrt{\kappa}+\sqrt{\theta}+\sqrt{D\theta}+\sqrt{D}\right)}{k(t)}\left(\sqrt{\frac{1}{\kappa}\left(D^2 + \frac{C^2}{\kappa\theta}\ln\left(\frac{\kappa T}{2\theta}+1\right)\right)} + 1\right).$$

$\square$

## D    PROOF OF THEOREM 5.1

*Proof of Theorem 5.1.* By the definition of dynamic regret, we have:

$$\text{DynamicReg}_i(T) = \sum_{t=1}^{T} \left( \max_{x \in \mathcal{X}_i} v_i(x, \pi_{-i}^t) - v_i(\pi^t) \right)$$

$$\leq \mathcal{O}(1) + \sum_{t=2}^{T} \sum_{i=1}^{N} \left( \max_{x \in \mathcal{X}_i} v_i(x, \pi_{-i}^t) - v_i(\pi^t) \right).$$

Here, we introduce the following lemma:

**Lemma D.1** (Lemma 2 of Cai et al. (2022a))**.** *For any $\pi \in \mathcal{X}$, we have:*

$$\sum_{i=1}^{N} \left( \max_{\tilde{\pi}_i \in \mathcal{X}_i} v_i(\tilde{\pi}_i, \pi_{-i}) - v_i(\pi) \right) \leq \text{GAP}(\pi) \leq D \cdot \max_{\tilde{\pi} \in \mathcal{X}} \langle V(\pi), \tilde{\pi} - \pi \rangle.$$

Therefore, we have:

$$\text{DynamicReg}_i(T) \leq \mathcal{O}(1) + \sum_{t=2}^{T} \text{GAP}(\pi^t).$$

Thus, from Theorem 4.1:

$$\text{DynamicReg}_i(T) \leq \mathcal{O}(1) + \sum_{t=2}^{T} \mathcal{O}\left( \frac{\ln T}{t} \right)$$

$$\leq \mathcal{O}\left( (\ln T)^2 \right).$$

$\square$

## E    EXPERIMENTAL DETAILS

### E.1    INFORMATION ON THE COMPUTER RESOURCES

The experiments were conducted on macOS Sonoma 14.4.1 with Apple M2 Max and 32GB RAM.

### E.2    HARD CONCAVE-CONVEX GAME

Following the setup in Ouyang & Xu (2021); Cai & Zheng (2023), we choose

$$A = \frac{1}{4} \begin{bmatrix} & & & -1 & 1 \\ & & \cdots & \cdots & \\ & -1 & 1 & & \\ -1 & 1 & & & \\ 1 & & & & \end{bmatrix} \in \mathbb{R}^{n \times n}, \quad b = \frac{1}{4} \begin{bmatrix} 1 \\ 1 \\ \cdots \\ 1 \\ 1 \end{bmatrix} \in \mathbb{R}^n, \quad h = \frac{1}{4} \begin{bmatrix} 0 \\ 0 \\ \cdots \\ 0 \\ 1 \end{bmatrix} \in \mathbb{R}^n,$$

and $H = 2A^\top A$.

### E.3    HYPERPARAMETERS

For each game, we carefully tuned the hyperparameters for each algorithm to ensure optimal performance. The specific parameters for each game and setting are summarized in Table 2.

| Game | Algorithm | $\eta$ | $T_\sigma$ | $\mu$ |
|---|---|---|---|---|
| Random Payoff (Full Feedback) | OG | 0.05 | - | - |
| | AOG | 0.05 | - | - |
| | APGA | 0.05 | 20 | 1.0 |
| | GABP | 0.05 | 10 | 1.0 |
| Random Payoff (Noisy Feedback) | OG | 0.001 | - | - |
| | AOG | 0.001 | - | - |
| | APGA | 0.001 | 2000 | 1.0 |
| | GABP | 0.001 | 1000 | 1.0 |
| Hard Concave-Convex (Full Feedback) | OG | 1.0 | - | - |
| | AOG | 1.0 | - | - |
| | APGA | 1.0 | 20 | 0.1 |
| | GABP | 1.0 | 20 | 0.1 |
| Hard Concave-Convex (Noisy Feedback) | OG | 0.5 | - | - |
| | AOG | 0.5 | - | - |
| | APGA | 0.5 | 50 | 0.1 |
| | GABP | 0.1 | 100 | 0.1 |

Table 2: Hyperparameters

# F  COMPARISON WITH EXISTING LEARNING ALGORITHMS

## F.1  RELATIONSHIP WITH ACCELERATED OPTIMISTIC GRADIENT ALGORITHM

Our GABP bears some relation to Accelerated Optimistic Gradient (AOG) (Cai & Zheng, 2023), which updates the strategy by:

$$\pi_i^{t+\frac{1}{2}} = \arg\max_{x \in \mathcal{X}_i} \left\{ \left\langle \eta \widehat{\nabla}_{\pi_i} v_i(\pi^{t-\frac{1}{2}}) + \frac{\pi_i^1 - \pi_i^t}{t+1}, x \right\rangle - \frac{1}{2} \left\| x - \pi_i^t \right\|^2 \right\},$$

$$\pi_i^{t+1} = \arg\max_{x \in \mathcal{X}_i} \left\{ \left\langle \eta \widehat{\nabla}_{\pi_i} v_i(\pi^{t+\frac{1}{2}}) + \frac{\pi_i^1 - \pi_i^t}{t+1}, x \right\rangle - \frac{1}{2} \left\| x - \pi_i^t \right\|^2 \right\}.$$

This can be equivalently written as:

$$\pi_i^{t+\frac{1}{2}} = \arg\max_{x \in \mathcal{X}_i} \left\{ \eta \left\langle \widehat{\nabla}_{\pi_i} v_i(\pi^{t-\frac{1}{2}}), x \right\rangle - \frac{1}{2} \left\| x - \frac{t\pi_i^t + \pi_i^1}{t+1} \right\|^2 \right\},$$

$$\pi_i^{t+1} = \arg\max_{x \in \mathcal{X}_i} \left\{ \eta \left\langle \widehat{\nabla}_{\pi_i} v_i(\pi^{t+\frac{1}{2}}), x \right\rangle - \frac{1}{2} \left\| x - \frac{t\pi_i^t + \pi_i^1}{t+1} \right\|^2 \right\}.$$

This means that AOG employs a convex combination $\frac{t\pi_i^t + \pi_i^1}{t+1}$ of the current strategy $\pi_i^t$ and initial strategy $\pi_i^1$ as the proximal point in gradient ascent. However, our GABP diverges from AOG in that it uses a convex combination $\frac{k(t)\sigma_i^{k(t)} + \sigma_i^1}{k(t)+1}$ of $\sigma_i^{k(t)}$ and $\sigma_i^1$ as the reference strategy for the perturbation term.

In terms of proof for a last-iterate convergence result, Cai & Zheng (2023) have employed the following potential function for the full feedback setting:

$$P^t := \frac{t(t+1)}{2} \left( \eta^2 \left\| -\widehat{V}(\pi^t) + c^t \right\|^2 + \eta^2 \left\| \widehat{V}(\pi^t) - \widehat{V}(\pi^{t-\frac{1}{2}}) \right\|^2 \right) + t\eta\langle -\widehat{V}(\pi^t) + c^t, \pi^t - \pi^1 \rangle,$$

where $c^t = \frac{\pi^{t-1} + \eta\widehat{V}(\pi^{t-\frac{1}{2}}) + \frac{1}{t}(\pi^1 - \pi^{t-1}) - \pi^t}{\eta}$ and $\widehat{V}(\cdot) = (\widehat{\nabla}_{\pi_i} v_i(\cdot))_{i \in [N]}$. Compared to our potential function $P^{k(t)}$, their potential function includes the term of $\eta^2 \left\| \widehat{V}(\pi^t) - \widehat{V}(\pi^{t-\frac{1}{2}}) \right\|^2$. In the noisy feedback setting, the value of this term could remain high even if $\pi^t = \pi^{t-\frac{1}{2}}$. Therefore, this term complicates providing a last-iterate convergence result for the noisy feedback setting. In contrast,

our potential function $P^{k(t)}$ does not contain the term depending on $\widehat{\nabla} v_i(\pi^t)$:

$$P^{k(t)} := \frac{k(t)(k(t)+1)}{2} \left\| \pi^{\mu,k(t)-1} - \hat{\sigma}^{k(t)-1} \right\|^2 \\ + k(t)(k(t)+1) \left\langle \hat{\sigma}^{k(t)} - \pi^{\mu,k(t)-1}, \pi^{\mu,k(t)-1} - \hat{\sigma}^{k(t)-1} \right\rangle.$$

This allows us to provide the last-iterate convergence rates both for full and noisy feedback settings.

### F.2 COMPARISON OF LAST-ITERATE CONVERGENCE RESULTS

In this section, we provide Table 3 for comparison with last-iterate convergence results of existing representative learning algorithms in monotone games.

| Algorithm | Full Feedback | Noisy Feedback |
|---|---|---|
| Extragradient (Cai et al., 2022a;b) | $\mathcal{O}(1/\sqrt{T})$ | N/A |
| Optimistic Gradient (Golowich et al., 2020b; Gorbunov et al., 2022; Cai et al., 2022a) | $\mathcal{O}(1/\sqrt{T})$ | N/A |
| Extra Anchored Gradient (Yoon & Ryu, 2021) | $\mathcal{O}(1/T)$ | N/A |
| Accelerated Optimistic Gradient (Cai & Zheng, 2023) | $\mathcal{O}(1/T)$ | N/A |
| Iterative Tikhonov Regularization (Koshal et al., 2010; Tatarenko & Kamgarpour, 2019) | N/A | Asymptotic* |
| Adaptively Perturbed Gradient Ascent (Abe et al., 2024) | $\tilde{\mathcal{O}}(1/\sqrt{T})$ | $\tilde{\mathcal{O}}(1/T^{\frac{1}{10}})$ |
| **Ours** | $\tilde{\mathcal{O}}(1/T)$ | $\tilde{\mathcal{O}}(1/T^{\frac{1}{7}})$ |

Table 3: Existing results on last-iterate convergence in monotone games. (*) means the result holds only under bandit feedback.

