# OpenReview forum: "Boosting Perturbed Gradient Ascent for Last-Iterate Convergence in Games"
_ICLR.cc/2025/Conference — ICLR 2025 Poster_

### Official Review · Reviewer_d7d7 · 2024-10-30

**Soundness:** 3
**Presentation:** 3
**Contribution:** 2
**Rating:** 6
**Confidence:** 3

**Summary:**

This paper proposes GABP, which addresses the online learning tasks in games that are monotone with respect to gradient. The GABP algorithm uses a perturbation on policy update such that the policy anchors on a linear combination of the initial policy and the selected anchor policy, which is shown to accelerate convergence and stabilize performance. The last-iterate convergence to NE is provided.

**Strengths:**

Using the carefully constructed perturbation, the newly proposed GABP achieves a faster convergence rate compared to APMD. A good contribution to the literature.

The related literature is detailed and explains the

**Weaknesses:**

The notations in this paper is a bit confusing to the readers, a subsection dedicated for notations could be helpful. In addition, the subscripts and superscripts in section 4 can be simplified, such as the stationary point as well as $\hat{\sigma}$.

Although Section E.1 provided a discussion between the proposed GABP with AOG, with a different motivation, etc. However, the exact update still appears similar to the reviewer.

The experiments do not demonstrate a clear advantage of GABP compared to existing methods listed as benchmark. Perhaps more explanation could be helpful on why in some cases certain methods fail and why GABP is better overall.

**Questions:**

See Weaknesses.

---

> ### Author Response · Authors · 2024-11-20
> **Response to Reviewer d7d7**
>
> We thank you for your positive feedback and constructive comments. The detailed answers to each of the questions can be found below.
>
> ---
>
> ### Weakness 1
>
> > The notations in this paper is a bit confusing to the readers, a subsection dedicated for notations could be helpful. In addition, the subscripts and superscripts in section 4 can be simplified, such as the stationary point as well as $\hat{\sigma}$.
> >
>
> ### Answer
>
> Thank you for your insightful suggestions! We have included a dedicated subsection for notations in Appendix G in our updated manuscript, providing a table to summarize and explain each notation we use. Additionally, we will strive to simplify the subscripts and superscripts in Section 4 as much as possible.
>
> ---
>
> ### Weakness 2
>
> > Although Section E.1 provided a discussion between the proposed GABP with AOG, with a different motivation, etc. However, the exact update still appears similar to the reviewer.
> >
>
> ### Answer
>
> The key difference between GABP and AOG lies in the introduction of the payoff perturbation, $\eta\mu\left(\pi^t - \frac{k(t)\sigma^{k(t)} + \sigma^1}{k(t)+1}\right)$, as mentioned in line 215. While this term may appear similar to the term AOG’s update formula, it serves a fundamentally different purpose. Specifically, AOG essentially modifies a proximal point from $\pi^t$ to $\frac{t\pi^t + \pi^1}{t+1}$ to stabilize the learning dynamics, as we have explained in lines 1599-1606. However, the term $\frac{\pi^1 - \pi^t}{t+1}$ is independent of the learning rate $\eta$ , and thus cannot be interpreted as a perturbation payoff. This means that **AOG cannot leverage the strong convexity of the (perturbed) payoff function**, and it has not been demonstrated to achieve last-iterate convergence in the noisy feedback setting.
>
> ---
>
> ### Weakness 3
>
> > The experiments do not demonstrate a clear advantage of GABP compared to existing methods listed as benchmark. Perhaps more explanation could be helpful on why in some cases certain methods fail and why GABP is better overall.
> >
>
> ### Answer
>
> Firstly, the primary advantage of GABP over APGA is its ability to **achieve faster convergence rates** (as shown in Theorems 4.1 and 4.3), even in challenging instances outside of the matrix game class. This is clearly observable in the results of the hard concave-convex game depicted in the two rightmost graphs of Figure 1.
>
> Secondly, with regard to AOG, as previously mentioned in our response to Weakness 2, the key advantage of GABP lies in its robustness in **achieving last-iterate convergence, regardless of the presence of noise**. This can be clearly seen in the 2nd and 4th graphs from the left in Figure 1.

---

> > ### Comment · Reviewer_d7d7 · 2024-12-02
> >
> > I have read the response and will keep my score.

---

### Official Review · Reviewer_gYTD · 2024-11-04

**Soundness:** 2
**Presentation:** 3
**Contribution:** 2
**Rating:** 5
**Confidence:** 3

**Summary:**

This paper introduces a novel algorithm, Gradient Ascent with Boosting Payoff Perturbation, designed to improve last-iterate convergence rates in monotone games. The research builds upon the existing framework of payoff perturbation methods, which aid in stabilizing convergence in multi-player games where strategy profiles evolve iteratively. The authors address challenges posed by noisy feedback in gradient-based learning environments and achieve enhanced convergence rates with a boosting approach that periodically re-initializes a reference strategy, thereby maintaining strong convexity in the payoff function.

**Strengths:**

* The GABP algorithm leverages a perturbation term to enhance the stability and convergence rate.
* Theoretical analysis covers both full and noisy feedback settings.
* The study includes diverse game settings and benchmarks against state-of-the-art algorithms.

**Weaknesses:**

* The proof strategy seems standard, and the technical contribution is not clarified in this paper.

**Questions:**

* It would be better to give more explanation on the intuition for the perturbation term (*) in Eq (3).
* Regarding the Hard Concave-Convex games in Appendix D.3, is there any reason that GABP uses a different step size (0.1) compared with other algorithms (0.5)?

---

> ### Author Response · Authors · 2024-11-20
> **Response to Reviewer gYTD**
>
> Thank you for taking the time to review our work and for providing your valuable feedback. The detailed answers to each of the questions can be found below.
>
> ---
>
> ### Weakness 1
>
> > The proof strategy seems standard, and the technical contribution is not clarified in this paper.
> >
>
> ### Answer
>
> We assert that our proof strategy is not standard. One of the technical contributions lies in developing the potential function $P^k$, which allows us to **derive faster convergence rates even for the noisy feedback setting**.
>
> Firstly, this innovation manages to derive a faster convergence rate $\mathcal{O}(1/k)$ of $\left\\|\pi^{\mu, \sigma^k} - \sigma^k\right\\|$, significantly surpassing the rate achieved by Abe et al. (2024). Secondly, although we did draw inspiration from the potential function used by Cai & Zheng (2023), our potential function is more versatile and adaptable. Cai & Zheng’s potential function is not directly applicable to the noisy feedback setting as they only focused on the full feedback setting. On the contrary, our potential function $P^k$ primarily depends on the anchoring strategy $\sigma^k$ and the stationary point $\pi^{\mu, \sigma^k}$, rather than the current strategy $\pi^t$, enabling us to derive the rate for the noisy feedback setting.
>
> ---
>
> ### Question 1
>
> > It would be better to give more explanation on the intuition for the perturbation term (*) in Eq (3).
> >
>
> ### Answer
>
> We would like to clarify that the primary purpose of presenting Eq. (3) is to underscore the deviation from APGA, thereby highlighting our GABP’s distinctive approach. In essence, the term (*) in Eq. (3) is intentionally designed to **facilitate a more gradual evolution of the anchoring strategy** than APGA. This is achieved by employing a convex combination of $\sigma^k$ and $\sigma^1$ as the anchoring strategy. We have provided a detailed explanation of this intuition in the equation in line 215 and further elaborated in lines 216-219.
>
> ---
>
> ### Question 2
>
> > Regarding the Hard Concave-Convex games in Appendix D.3, is there any reason that GABP uses a different step size (0.1) compared with other algorithms (0.5)?
> >
>
> ### Answer
>
> As stated in Appendix D.3, Table 1 reports the hyperparameter settings in which each algorithm achieves its best performance for each instance. More specifically, we presented the experimental results with the optimal learning rate chosen from the set of $\\{1.0, ~0.5, ~0.2, ~0.1, ~0.05, ~0.02, ~0.01\\}$. The step size of $0.1$ for the GABP was found to be the most effective in our experiments; hence, it was used in comparison to the other algorithms, which were found to perform best with a step size of $0.5$.

---

### Official Review · Reviewer_K3ym · 2024-11-04

**Soundness:** 4
**Presentation:** 3
**Contribution:** 3
**Rating:** 8
**Confidence:** 2

**Summary:**

The paper focuses on last-iterate convergence on smooth, monotone games for both the full feedback and noisy feedback settings. The authors propose a novel payoff perturbation term that dynamically interpolates the current anchoring strategy and the initial anchoring strategy. The interpolation achieves a trade-off between convergence speed and stability to gradient noise. The authors proved that the proposed algorithm achieves state-of-the-art last-iterate convergence rates for both the exact and noisy gradient feedback settings. Through experiments, it’s shown that the proposed algorithm achieves comparable or superior convergence speed in random payoff and hard concave-convex games, in noiseless and noisy gradient settings.

**Strengths:**

- The technical contribution is strong. The proposed GABP algorithm is well motivated and explained, and the authors have shown proof for the last iterate convergence rates.
- The paper is well-written. Proof sketches are shown for main theorems.

**Weaknesses:**

The paper does not discuss limitations.

**Questions:**

- In Figure 1, why is GAP shown for the random payoff game and the $r^{tan}$ shown for the hard concave-convex game? Can you show the other metric for these games as well?

---

> ### Author Response · Authors · 2024-11-20
> **Response to Reviewer K3ym**
>
> We thank you for your positive feedback and constructive comments. Your question is answered below.
>
> ---
>
> ### Question 1
>
> > In Figure 1, why is GAP shown for the random payoff game and the $r^{tan}$ shown for the hard concave-convex game? Can you show the other metric for these games as well?
> >
>
> ### Answer
>
> We just decided to present the metrics that have been commonly used in the literature for each respective game. For the random payoff game, we utilized the gap function as our metric, following the experiments conducted by Abe et al. (2024). Similarly, for the hard concave-convex game, we used the tangent residual, following Cai & Zheng (2023).
>
> We appreciate your suggestion for a more comprehensive comparison. To address this, we have included an additional figure in Appendix F of our updated manuscript for comparison of last-iterate convergence rates in terms of the gap function for all instances.

---

### Official Review · Reviewer_tqic · 2024-11-04

**Soundness:** 3
**Presentation:** 2
**Contribution:** 2
**Rating:** 6
**Confidence:** 3

**Summary:**

This paper considers multi-player monotone games and proposes a perturbed gradient ascent algorithm with improved performance in both deterministic and stochastic settings. Numerical simulations are provided.

**Strengths:**

The paper is well-organized and well-written. The proposed algorithm of Gradient Ascent with Boosting Payoff Perturbation is easy to implement and enjoys strong convergence guarantees. The numerical simulations are extensive and demonstrate the empirical performance of the proposed algorithm in various settings.

**Weaknesses:**

(1) Algorithmically, there is only one difference between the proposed algorithm and the Adaptively Perturbed Mirror Descent (APMD) from Abe et al. (2024): instead of directly using $\sigma_i^k$ as the anchoring strategy, a convex combination of $\sigma_i^k$ and $\sigma_i^1$ is used as the anchoring strategy. Surprisingly, with this seemingly minor modification, the resulting algorithms achieve improved performance guarantees. I did not quite follow why such a modification results in improved performance, despite the illustration from line 216 to line 225. What is the intuition behind this modification? Why does it help improve the performance guarantees? Are there intuitive and technical explanations behind it?

(2) In terms of the analysis, are there any major technical challenges in extending the approach from Abe et al. (2024) to that of this work? If so, how did the authors overcome it?

**Questions:**

(1) According to the definition, $\sigma_i^1=\pi^1$. Am I missing anything?

(2) Why are we not using the Nash Gap as a metric, which seems more natural?

---

> ### Author Response · Authors · 2024-11-20
> **Response to Reviewer tqic**
>
> We thank you for your positive feedback and constructive comments. The detailed answers to each of the questions can be found below.
>
> ---
>
> ### Weakness 1
>
> > (1) Algorithmically, there is only one difference between the proposed algorithm and the Adaptively Perturbed Mirror Descent (APMD) from Abe et al. (2024): instead of directly using $\sigma_i^k$ as the anchoring strategy, a convex combination of $\sigma_i^k$ and $\sigma_i^1$ is used as the anchoring strategy. Surprisingly, with this seemingly minor modification, the resulting algorithms achieve improved performance guarantees. I did not quite follow why such a modification results in improved performance, despite the illustration from line 216 to line 225. What is the intuition behind this modification? Why does it help improve the performance guarantees? Are there intuitive and technical explanations behind it?
> >
>
> ### Answer
>
> APMD and GABP both use anchoring strategies to determine the magnitude of perturbations, but they differ in how this magnitude is computed. APMD specifies the perturbation magnitude as the distance between the current strategy $\pi^t$ and the anchoring strategy $\sigma^k$. In contrast, GABP computes this distance in a more intricate way, considering the distance between the current strategy and a convex combination of the current anchoring strategy $\sigma^k$ and the initial anchoring strategy $\sigma^1$.
>
> The purpose of using this convex combination is to **ensure that the updated strategy does not deviate too far from the current anchoring strategy**, much like how the regularizer functions in MD or FTRL. This approach mitigates large changes or fluctuations in the updated strategies, which can destabilize the dynamics and hinder convergence. Consequently, GABP facilitates a more gradual evolution of the anchoring strategy, starting from $\sigma^1$, and achieves more stable learning dynamics with an improved convergence rate compared to APMD.
>
> ---
>
> ### Weakness 2
>
> > (2) In terms of the analysis, are there any major technical challenges in extending the approach from Abe et al. (2024) to that of this work? If so, how did the authors overcome it?
> >
>
> ### Answer
>
> One of the key technical challenges we faced was improving the convergence rate in terms of the distance between the $k$-th anchoring strategy $\sigma^k$  and the corresponding current strategy $\pi^{\mu,\sigma^k}$, i.e., $\left\\|\pi^{\mu, \sigma^k} - \sigma^k\right\\|$, as compared to Abe et al. (2024). In fact, we have successfully improved the rate from $\mathcal{O}(1/\sqrt{k})$ to $\mathcal{O}(1/k)$, where $k$ indicates the number of times the anchoring strategy is updated.
>
> Cai and Zheng (2023) achieved a rate of $\mathcal{O}(1/t)$ under full feedback, where $t$ represents the number of iterations. They introduced the concept of a non-increasing **potential function** to derive that rate **only under full feedback**. Our analysis follows a similar approach to them, but their potential function fails short under noisy feedback. To achieve **last-iterate convergence under noisy feedback**, we developed a novel potential function, referred to as $P^k$, which incorporates the anchoring strategy in a non-trivial way. It's important to note that this is not a mere extension of Cai and Zheng (2023). The details are observed at the beginning of Section 4.3 and Appendix E.1. We will highlight the technical advancements compared to Abe et al. (2024) and Cai and Zheng (2023) accordingly.
>
> ---
>
> ### Question 1
>
> > (1) According to the definition, $\sigma_i^1=\pi^1$. Am I missing anything?
> >
>
> ### Answer
>
> No. As you pointed out, for our experiments, we have just implemented our algorithm such that the current and anchoring strategies are the same, i.e.,  $\sigma^1=\pi^1$. However, we would like to note that we can initialize them independently, allowing the anchoring strategy to differ from the initial strategy, i.e., $\sigma^1\neq\pi^1$. Even in such cases, we still achieve the same convergence rate as stated in Theorems 4.1 and 4.3.
>
> ---
>
> ### Question 2
>
> > (2) Why are we not using the Nash Gap as a metric, which seems more natural?
> >
>
> ### Answer
>
> We can argue that the Nash Gap is **a weaker measure** than the gap function and the tangent residual. This argument is substantiated by the following inequality, derived under the monotonicity assumption:
>
> $$
> \begin{aligned}
> \mathrm{NashGap}(\pi) &= \sum\_{i\in [N]} \max\_{\tilde{\pi}\_i\in \mathcal{X}\_i}\left(v\_i(\tilde{\pi}\_i, \pi\_{-i}) - v_i(\pi)\right) \\\\
> &\leq \sum\_{i\in [N]}\max\_{\tilde{\pi}\_i\in \mathcal{X}\_i}\langle \nabla_{\pi\_i}v\_i(\pi), \tilde{\pi}\_i - \pi\_i\rangle = \mathrm{GAP}(\pi).
> \end{aligned}
> $$
>
> This inequality implies that the Nash Gap is upper bounded by the gap function and the tangent residual. Furthermore, these metrics are commonly employed in the literature on monotone games, which are beyond simple matrix games [Cai & Zheng, 2023; Abe et al., 2024].

---

### Meta-Review · Area_Chair_6WiD · 2024-12-21

**Metareview:**

The paper introduces a modifed perturbed scheme called GABP and show last iterate convergence with rate of 1/T and 1/T^1/7 for full feedback and noisy feedback models on monotone games. The reviewers agree that the paper has merits/novelty and all of them were positive or slightly positive about the contributions. We recommend a weak acceptance of this paper.

**Additional Comments On Reviewer Discussion:**

The reviewers were positive about this work and remained positive after the rebuttal.

---

### Decision · Program_Chairs · 2025-01-22

Accept (Poster)